# LONG-TERM TIME SERIES FORECASTING WITH VISION TRANSFORMER

## ABSTRACT

Transformer has been widely used for modeling sequential data in recent years. For example the Vision Transformer (ViT), which divides an image into a sequence of patches and uses Transformer to discover the underlying correlations between the patches, has become particularly popular in Computer Vision. Considering the similarity of data structure between time series data and image patches, it is reasonable to apply ViT or its variations for modeling time series data. In this work, we explore this possibility and propose the Swin4TS algorithm. It incorporates the window-based attention and hierarchical representation techniques from the Swin Transformer, a well-known ViT algorithm, and applies them to the long-term forecasting of time series data. The window-based attention enables the algorithm to achieve linear computational complexity, while the hierarchical architecture allows the representation on various scales. Furthermore, Swin4TS can flexibly adapt to channel-dependence and channel-independence strategies, in which the former can simultaneously capture correlations in both the channel and time dimensions, and the latter shows high training efficiency for large datasets. Swin4TS outperforms the latest baselines and achieves state-of-the-art performance on 8 benchmark datasets. More importantly, our results demonstrate the potential of transferring the Transformer architecture from other domains to time series analysis, which enables research on time series to leverage advancements at the forefront of other domains.

## 1 INTRODUCTION

Long-term Time Series Forecasting (LTSF) provides crucial support and guidance for various domains such as intelligent transportation (Li & Zhu, 2021; Rao et al., 2022), smart manufacturing (Zi et al., 2021; Wang et al., 2022b), and healthcare (Wang et al., 2022a; Shokouhifar & Ranjbarimesan, 2022), and it poses significant challenges due to complex long-term dependencies, intricate interplay of multiple variables, and other domain-related issues. Recently, deep learning methods, particularly Transformer-based approaches, have received significant attention for LTSF. A succession of algorithms have emerged, such as Informer (Zhou et al., 2021), Autoformer (Wu et al., 2021), FEDformer (Zhou et al., 2022), and PatchTST (Nie et al., 2023). These algorithms have been elaborately designed to either focus on the characteristics of time series or the quadratic computational complexity of Transformer, and have exhibited gradually improved performance on benchmark datasets.

Transformer is designed originally for modeling word sequences, and now it has become the most mainstream model in the domain of Natural Language Processing (NLP). Another example of Transformer's successful application is the Vision Transformer (ViT), which has become a hot topic in the research of Computer Vision (CV). Despite images not being sequential data, ViT divides them into small image patches (as shown in Fig. 1), which form the image sequences and further enable the application of Transformer to extract spatial features. From a perspective of data structure, both the ensemble of words, image patches, or time variables, are shown as sequential data and thus can be effectively modeled using Transformer. In particular, image- and time-sequences share two common characteristics. Firstly, their lengths are typically fixed. For example, in the image classification tasks, the scale of each image is usually consistent, and in time series prediction tasks, the length of the historical series is also fixed. However, NLP tasks often have variable input lengths for word sequences. Additionally, both image- and time-sequences require pre-defined scales for attention, for example, times series also require division into patches to eliminate the randomness

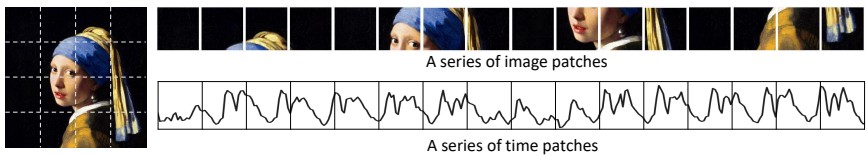

Figure 1: Illustrations of a series of image patches and a series of time patches.

and enhance the representation, whereas the scale of word sequences is inherently determined by individual words.

*This raises a natural question: can advanced ViT models be applied to facilitate time series modeling?* If the answer is "Yes", connecting time series with ViT would allow leveraging the fruitful well-established models of ViT for time series analysis. Very recent studies have showcased some heuristic examples toward this hypothesis, such as MV-DTSA (Yang et al., 2023) and ViTST (Li et al., 2023) which draw time sequences into binary images and then utilize mature CV models for downstream forecasting and classification tasks, respectively. However, these methods do not directly align time series with CV model on the data structure. Besides, converting the time series into images in this way might introduce redundant information.

In this article, we propose the Swin Transformer for Time Series (Swin4TS) algorithm to tackle LTSF, as a response to the aforementioned hypothesis. Swin Transformer (Liu et al., 2021c) is a well-known variant of ViT, and here we borrow two core designs from it: window-based attention and hierarchical representation. A major drawback of Transformer-based methods is their quadratic computational complexity with input time series. Swin4TS overcomes this by restricting attention to a size-fixed window, resulting in linear complexity and enabling the processing of longer inputs. Additionally, the hierarchical representation captures temporal information at different scales, leading to a more comprehensive representation of time series data. Benefiting from the similarity in data structure between time series and image patches, Swin4TS can easily adapt to channel-dependence and channel-independence strategies, where the former considers the multivariate correlation, while the latter does not. These two strategies complement each other in different cases, enabling Swin4TS to achieve the best performance compared to the latest baselines.

Our contributions with Swin4TS are as follows:

(1) We propose the Swin4TS algorithm for LTSF, motivated by the similarity of data structure between time series and image patches. Swin4TS has linear computational complexity and allows representation across multiple scales. Besides, it is designed to be compatible to either channel-dependence and channel-independence strategies, which consider the multivariate correlation or not, respectively.

(2) We evaluate Swin4TS on 32 prediction tasks across 8 benchmark datasets and achieve performance surpassing the latest baseline methods (both Transformer-based and non-Transformer-based) on almost all tasks.

(3) We successfully apply techniques from ViT to LTSF, indicating the feasibility of modeling time series modality using architectures from image modality. This allows advancements at the forefront of ViTs to facilitate research in time series analysis.

## 2 RELATED WORK

**Transformers for LTSF** Transformer model excels in processing sequential data, which has been extensively validated in the domain of NLP (Vaswani et al., 2017). Therefore, it has recently attracted growing interests in time series analysis. Notable works include Informer (Zhou et al., 2021), which proposes a sparse attention mechanism that only considers the top-ranked attention components; Pyraformer (Liu et al., 2021a) and Triformer (Cirstea et al., 2022) aim to reduce the computational complexity by introducing a pyramid or triangular structure to hierarchically perform attention; Autoformer (Wu et al., 2021) and FEDformer (Zhou et al., 2022) priorly decompose the time series to obtain key temporal information that enhances the prediction accuracy; Crossformer (Zhang & Yan, 2023) explicitly considers the multivariate correlations instead of fusing variables

by an embedding layer; PatchTST (Nie et al., 2023) divides the time series into patches to reduce complexity and models the time series independently. Scaleformer (Shabani et al., 2023) uses an external hierarchy design to obtain predictions at multiple scales and is compatible with various transformer-based models.

**LTSF with cross-domain models** Transformer exhibits the ability to adapt to a wide range of domains. A fascinating topic revolves around whether Transformer has a universal representation ability that allows for transferring knowledge between different domains. Recently, Zhou et al. (Zhou et al., 2023) proposed utilizing the Frozen Pretrained Transformer technique to fine-tune the language model GPT2 for handling temporal-related tasks such as forecasting, anomaly detection, classification, and imputation. Surprisingly, they achieved state-of-the-art performance on all tasks except forecasting the second best. In addition to fine-tuning, another approach for cross-domain temporal processing is to transform the temporal data into the content of another domain, and directly employ mature models from that domain for modeling. For instance, MV-DTSA (Yang et al., 2023) draws time series to the binary images and utilizes the U-net (Falk et al., 2018) or DeepLabV2 (Chen et al., 2016) (popular generative CV models) to generate the predicted images which depict the change of time series. Similarly, ViTST (Li et al., 2023) draws irregular times series into images and then uses Swin Transformer to mine the underlying patterns for downstream classification tasks.

**Vision Transformer** ViT is a hotspot of current CV research, and it has spawned a series of models Yuan et al. (2021); Wang et al. (2021); Chen et al. (2021) for various downstream tasks in CV. Its core principle is to divide the image into a series of image patches and convert these patches into sequential data, which is then processed by Transformer. Typical models include Swin Transformer (Liu et al., 2021c;b;d) and Transformer in Transformer (TNT) (Han et al., 2021). Swin Transformer considers space correlations in images by introducing the window-based attention and hierarchical design, which performs well in processing large-size images. It has achieved leading performance on multiple image classification benchmark datasets and also has remarkable advantages in computational efficiency. TNT introduces another ViT model on top of the existing ViT model to consider fine-grained information. These two ViT models separately process the patches at two different scales, and their representations are fused for downstream tasks. Despite both the TNT and Swin Transformer adopting a hierarchical structure, the computational complexity in practice of the TNT is higher than that of the Swin Transformer.

## 3 METHODOLOGY

### 3.1 PROBLEM DEFINITION

The time series forecasting problem can be described as: given a look-back window $L : \mathbf{X} = (\mathbf{x}_1, ..., \mathbf{x}_L) \in \mathbb{R}^{M \times L}$ of multivariate time series with $M$ channels (variables), where each $\mathbf{x}_t$ at time step $t$ is a $M$-dimensional vector, the goal is to forecast the future $T$ values $(\mathbf{x}_{L+1}, .., \mathbf{x}_{L+T}) \in \mathbb{R}^{M \times T}$. The real values of future series are depicted as $(\mathbf{y}_{L+1}, .., \mathbf{y}_{L+T}) \in \mathbb{R}^{M \times T}$, thus one can get the error between forecasting values and real values to measure the performance of forecasting.

For the modeling of multi-channel time series, it can be divided into two strategies: channel-dependence (CD) and channel-independence (CI), resulting in two variants Swin4TS/CD and Swin4TS/CI, respectively. The former considers the multivariate correlation, while the latter does not.

### 3.2 MODEL ARCHITECTURE

**Channel-independence strategy** We first consider the Swin4TS/CI with CI strategy, namely $X^m = (x_1^m, \ldots, x_L^m) \in \mathbb{R}^{1 \times L}$ as the series $m \in [1, M]$ is independently considered. For simplicity, we omit the superscript $m$ in what follows. Swin4TS/CI essentially shares the same network across all channels. Thus, all channels during training are trained together to obtain a generalized model.

**Window-based attention** A significant drawback of Transformer-based models is the complexity of $O(L^2)$ for encoding time series of length $L$. In this work, we adopt Window-based Attention (WA) to achieve linear complexity. As shown in Fig. 2(a), an independent time series $X$ is first divided into $W$ windows $\{x^i\}$, $i \in [1, W]$. Then each window $x^i$ is further divided into $N$ patches

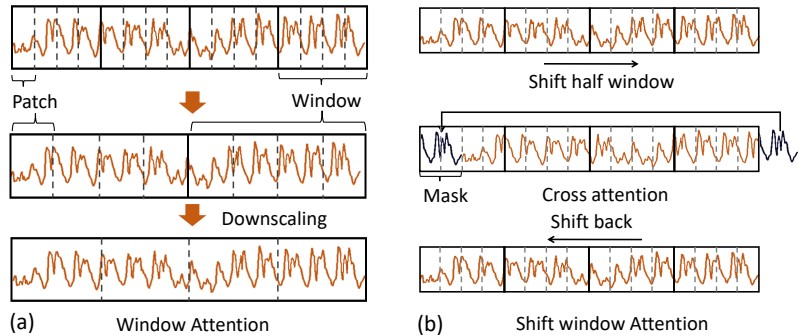

Figure 2: (a) Mechanism of window attention and hierarchical design. The solid and dashed lines distinguish the windows and patches, respectively. Attention among patches is localized within a window. (b) Mechanism of shift-window attention for a time series.

with patch length $P$, resulting in $\mathbf{x}^i \in \mathbb{R}^{N \times P}$ ($L = W \times N \times P$). For $j$-th patch $x_j^i$ in $i$-th window,

$$x_j^i = \{x_{t,j}^i \mid (i-1)NP + (j-1)P < t < (i-1)NP + jP\},\, j \in [1, N], i \in [1, W]. \quad (1)$$

Unlike PatchTST and Crossformer, attention here is applied within each window to focus on local interactions. Thus the overall computational complexity is $O(L/NP \cdot N^2)$, and it actually is $O(L)$ due to that $N$ and $P$ are fixed within each window. For $M$ independent channels, it further comes to $O(ML)$.

$\mathbf{x}^i$ is first embedded to the latent space with dimension $D$ by a learnable linear projection matrix $\mathbf{W}_p \in \mathbb{R}^{P \times D}$, i.e., $\mathbf{z}^i = \mathbf{x}^i \mathbf{W}_p \in \mathbb{R}^{N \times D}$. Then $\mathbf{z}^i$ is mapped to a sequence of output $\mathbf{O} = \{\mathbf{O}_h\}$ by:

$$\mathbf{O}_h = \text{Softmax}(\mathbf{Q}_h \mathbf{K}_h^\top / \sqrt{d}) \mathbf{V}_h \quad (2)$$

where $\mathbf{O}_h \in \mathbb{R}^{N \times D/H}$, and $h \in [1, H]$ denotes the one of $H$ heads. Here $\mathbf{Q}_h = \mathbf{z}^i \mathbf{W}_h^Q$, $\mathbf{K}_h = \mathbf{z}^i \mathbf{W}_h^K$, $\mathbf{V}_h = \mathbf{z}^i \mathbf{W}_h^V$ are projected queries, keys and values corresponding to the head $h$ with learnable projection matrix $\mathbf{W}_h^Q \in \mathbb{R}^{D \times d}$, $\mathbf{W}_h^K \in \mathbb{R}^{D \times d}$, $\mathbf{W}_h^V \in \mathbb{R}^{D \times D/H}$, respectively.

To consider the relation between two consecutive windows, we refer to the Shift-Window Attention (SWA) used in the Swin Transformer for image processing, and make it applicable for time series processing. In Fig. 2(b), the representation of time series is shifted half of window to the right, and the part sticking out from the right side complements the leftmost window. Then the new representation is again processed by window-based attention (note the nonphysical part in the first window is masked), ending with shifting it back to the previous position.

A complete attention module additionally includes the Batch Normalization (BN) layers [1] and a Feed Forward (FF) layer with residual connections (as illustrated in Fig. 3). Overall, the attention process can be described as:

$$\mathbf{z}_w^i = \text{BN}(\text{WA}(\mathbf{z}^i) + \mathbf{z}^i) \quad (3)$$

$$\mathbf{z}^i = \text{BN}(\text{FF}(\mathbf{z}_w^i) + \mathbf{z}_w^i) \quad (4)$$

$$\mathbf{z}_s^i = \text{BN}(\text{SWA}(\mathbf{z}^i) + \mathbf{z}^i) \quad (5)$$

$$\mathbf{z}^i = \text{BN}(\text{FF}(\mathbf{z}_s^i) + \mathbf{z}_s^i) \quad (6)$$

where $\mathbf{z}^i \in \mathbb{R}^{N \times D}$ that preserves the shape as the input. Finally, all the windows' output can be merged as $\mathbf{Z} = \{\mathbf{z}^i\} \in \mathbb{R}^{W \times N \times D}$, $i \in [1, W]$.

**Hierarchical representation** To this end, the model has only focused on a single scale, but information from other scales can represent the time series more comprehensively. Therefore, we introduce the hierarchical representation. For ease of description, we firstly consider two scales that correspond to local and global scales, respectively.

---

[1]LayerNorm is usually adopted for ViT, but it is shown (Zerveas et al., 2020) that BatchNorm outperforms LayerNorm in Transformers for modeling time series.

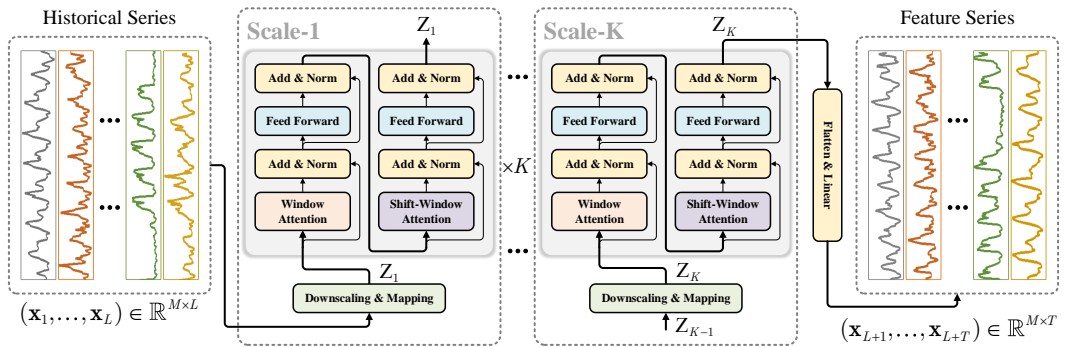

Figure 3: Architecture of Swin4TS with K-stage hierarchical design. For multivariate forecasting as illustrated, the inputs and outputs are $M$ historical series with length $L$ and $T$, respectively. For univariate forecasting, the inputs are 1 and $M$ historical series for CI and CD strategies, respectively.

As described above, the input $X$ is partitioned into $W$ windows $\{x^i\}$ which subsequently transfer to $\mathbf{Z}_{local} = \{\mathbf{z}^i\} \in \mathbb{R}^{W \times N \times D}$, representing the local information. Then $\mathbf{Z}_{local}$ is downscaled to obtain

$$\mathbf{Z}_{global} = \mathrm{DS}(\mathbf{Z}_{local}) \in \mathbb{R}^{W/2 \times N \times 2D} \tag{7}$$

where $\mathrm{DS}(\cdot)$ consists of the unfold and reshape operations followed by a linear layer to map the target dimension. $\mathbf{Z}_{global}$ contains half of windows with invariable patch number $N$, thus it represents a larger scale compared to $\mathbf{Z}_{local}$. Similarly, $\mathbf{Z}_{global}$ also needs to undergo (shift) window-based attention to extract global scale information. From a general view, as illustrated in Fig. 3, one can consider $K$ scales for a time series. For the consecutive $k^{\text{th}}$ and $k+1^{\text{th}}$ scales,

$$\mathbf{Z}_{k+1} = \mathrm{Attention}(\mathrm{DS}(\mathbf{Z}_k)), \ k \in [1, K-1]. \tag{8}$$

After obtaining the output of the last scale $\mathbf{Z}_K$, the final prediction $(x_{L+1}, .., x_{L+T})$ can be obtained through a flatten layer with a linear head.

**Channel-dependence strategy** The above section describes the implementation of Swin4TS with the CI strategy. In fact, Swin4TS can be easily extended to the CD strategy to simultaneously consider correlations in both the time and channel dimensions, known as Swin4TS/CD. Under the CD strategy, multivariate time series data $\mathbf{X}$ is structurally similar to an image, allowing us to also apply window-based attention and hierarchical representation in the channel dimension. Concretely, $\mathbf{X}$ is first divided equally into $W_c \times W_t$ windows, resulting in $\{\mathbf{x}^i\}$, $i \in [1, W_c W_t]$, where the subscripts $c$ and $t$ correspond to channel and time, respectively. For each window $\mathbf{x}^i$, it is further divided equally at the channel and time dimension and obtains $N_c \times N_t$ patches with height $P_c$ and width $P_t$, namely $\mathbf{x}^i \in \mathbb{R}^{N_c N_t \times P_c P_t}$.

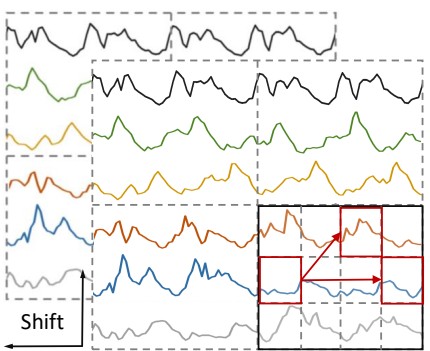

Figure 4: Schematics of Swin4TS with CD strategies. The black and red boxes respectively represent the window and patch. Shift operations are conducted along channel and time dimensions as shown by the arrows.

It should be noted that although the SWA used to consider cross-window attention is consistent with the CI strategy, it comes slightly more complex since the shift occurs in both the channel and time dimensions, as shown in Fig. 4. Additionally, in the hierarchical representation, window downscaling needs to be performed separately in both two dimensions. Apart from these considerations, the CD and CI strategies are consistent in the frame of Swin4TS.

When processing long multi-variable sequences for prediction, the design of last Linear layer as CI strategy may not be a good choice. Additionally, we offer a possible solution as an alternative design (shown in Appendix D).

Compared to the CI strategy, a patch here not only correlates to the patches belonging to the same channel, as can be seen in Fig. 4, but also correlates to patches across channels. Moreover, through

the SWA and hierarchical operations, it becomes possible to establish correlations with channels across windows.

**Loss function** Following the common setting, we utilize the Mean Squared Error (MSE) loss as a metric to quantify the disparity between the predicted values and the ground truth. The loss for each channel is collected and then averaged across $T$ predicted time steps and all channels to obtain the overall objective loss:

$$\mathcal{L} = \frac{1}{TM} \Sigma_{m=1}^{M} \Sigma_{t=1}^{T} ||\mathbf{x}_{L+t} - \mathbf{y}_{L+t}||^2. \tag{9}$$

## 4 EXPERIMENTS

### 4.1 EXPERIMENTAL SETUP

**Datasets** We evaluate the performance of our proposed Swin4TS on 8 real-world public benchmark datasets, including Weather, Traffic, Electricity, ILI and 4 ETT datasets (ETTh1, ETTh2, ETTm1 and ETTm2). More details can be referred to Appendix A.1.

**Settings** The length $L$ of historical series is set to 108 for ILI and 512 for the others, and four prediction tasks are considered with prediction lengths $T \in \{24, 36, 48, 60\}$ for ILI and $T \in \{96, 192, 336, 720\}$ for the others. Mean Squared Error (MSE) and Mean Absolute Error (MAE) are utilized as evaluation metrics. For more algorithmic details, hyperparameter settings, and information about the training, please refer to Appendix A.2

**Compared baselines** We compare our proposed method with 7 the most recent and popular models in long-term forecasting tasks, including 4 Transformer-based model: PatchTST (Nie et al., 2023), Crossformer (Zhang & Yan, 2023), FEDformer (Zhou et al., 2022) and Autoformer (Wu et al., 2021); 4 none-Transformer-based model: DLinear (Zeng et al., 2023), MICN (Wang et al., 2023) , TimesNet (Wu et al., 2023) and N-HiTS (Challu et al., 2023). Besides, PatchTST and DLinear employ the CI strategy, while the remaining models adopt the CD strategy. In these models, PatchTST and DLinear use longer historical series with $L = 336$ or 512 as input. On the other hand, the remaining models use shorter historical series with $L = 96$ as input. We know that different models require suited $L$ to achieve their best performance. To make a fair comparison, for each dataset, the involved baseline algorithms are evaluated with $L = 96$, 336, and 512 respectively, and then the best one is selected as their final result (see Appendix B.3). This ensures that we always compare with the strongest results of each baseline algorithm.

### 4.2 RESULTS

**Main results** Table 1 presents the prediction results of Swin4TS/CI and Swin4TS/CD compared with other baselines on 8 datasets. Overall, Swin4TS achieves the state-of-the-art performance across all datasets (full results can be seen in Appendix B.1). Particularly, on the ILI and Traffic datasets, Swin4TS/CD and Swin4TS/CI outperform the previous best results by 15.8% (1.967→1.657) and 10.3% (0.397→0.356) respectively. On the 4 ETT datasets, both Swin4TS/CI

Table 1: Multivariate long-term series forecasting on 8 datasets. Bold/underline indicates the best/second. Algorithm with * suggests the use of CI strategy otherwise the CD strategy. All the results are averaged from 4 different prediction lengths.

| Models | Swin4TS/CI* (ours) | | Swin4TS/CD* (ours) | | PatchTST/64* (2023) | | DLinear* (2023) | | MICN (2023) | | Crossformer (2023) | | TimesNet (2023) | | N-HiTS (2023) | | FEDformer (2022) | | Autoformer 2021 | |
|---|---|---|---|---|---|---|---|---|---|---|---|---|---|---|---|---|---|---|---|---|
| Metrics | MSE | MAE | MSE | MAE | MSE | MAE | MSE | MAE | MSE | MAE | MSE | MAE | MSE | MAE | MSE | MAE | MSE | MAE | MSE | MAE |
| Weather | **0.220** | **0.259** | 0.224 | 0.260 | 0.224 | 0.262 | 0.246 | 0.300 | 0.241 | 0.295 | 0.250 | 0.286 | 0.225 | 0.286 | 0.249 | 0.274 | 0.310 | 0.357 | 0.335 | 0.379 |
| Traffic | **0.356** | **0.254** | 0.526 | 0.336 | 0.397 | 0.270 | 0.434 | 0.295 | 0.484 | 0.306 | 0.523 | 0.290 | 0.620 | 0.336 | 0.452 | 0.311 | 0.604 | 0.372 | 0.617 | 0.384 |
| Electricity | **0.157** | **0.250** | 0.176 | 0.280 | 0.160 | 0.254 | 0.166 | 0.264 | 0.178 | 0.287 | 0.194 | 0.291 | 0.193 | 0.295 | 0.186 | 0.287 | 0.207 | 0.321 | 0.214 | 0.327 |
| ILI | 1.740 | 0.874 | **1.657** | **0.849** | 1.967 | 0.921 | 2.169 | 1.041 | 2.548 | 1.105 | 3.322 | 1.222 | 2.139 | 0.931 | 2.051 | 0.926 | 2.597 | 1.070 | 2.819 | 1.120 |
| ETTh1 | 0.411 | 0.428 | **0.406** | **0.425** | 0.419 | 0.438 | 0.423 | 0.437 | 0.500 | 0.500 | 0.454 | 0.471 | 0.458 | 0.450 | 0.434 | 0.467 | 0.428 | 0.454 | 0.473 | 0.477 |
| ETTh2 | 0.337 | 0.385 | **0.335** | **0.383** | 0.342 | 0.388 | 0.431 | 0.447 | 0.523 | 0.492 | 1.023 | 0.729 | 0.410 | 0.442 | 0.409 | 0.432 | 0.388 | 0.434 | 0.422 | 0.443 |
| ETTm1 | **0.341** | **0.376** | 0.348 | 0.380 | 0.353 | 0.382 | 0.357 | 0.379 | 0.371 | 0.398 | 0.435 | 0.450 | 0.400 | 0.406 | 0.362 | 0.394 | 0.382 | 0.422 | 0.515 | 0.493 |
| ETTm2 | 0.250 | 0.311 | **0.248** | **0.308** | 0.255 | 0.315 | 0.267 | 0.332 | 0.291 | 0.351 | 0.908 | 0.667 | 0.282 | 0.334 | 0.279 | 0.330 | 0.292 | 0.343 | 0.310 | 0.357 |

Table 2: Univariate long-term forecasting results with Swin4TS. Bold/underline indicates the best/second. Algorithm with * suggests the use of CI strategy otherwise the CD strategy. All the results are averaged from 4 different prediction lengths.

| Models | Swin4TS/CI* (ours) | | Swin4TS/CD (ours) | | PatchTST/64* (2023) | | DLinear* (2023) | | MICN (2023) | | TimesNet (2023) | | FEDformer (2022) | | Autoformer (2021) | |
|---|---|---|---|---|---|---|---|---|---|---|---|---|---|---|---|---|
| Metrics | MSE | MAE | MSE | MAE | MSE | MAE | MSE | MAE | MSE | MAE | MSE | MAE | MSE | MAE | MSE | MAE |
| ETTh1 | 0.073 | 0.214 | **0.069** | **0.211** | 0.074 | 0.215 | 0.104 | 0.247 | 0.122 | 0.274 | 0.086 | 0.231 | 0.111 | 0.257 | 0.105 | 0.252 |
| ETTh2 | **0.160** | **0.319** | 0.175 | 0.344 | 0.177 | 0.335 | 0.198 | 0.350 | 0.207 | 0.360 | 0.196 | 0.353 | 0.206 | 0.350 | 0.218 | 0.364 |
| ETTm1 | **0.047** | **0.162** | 0.055 | 0.181 | 0.048 | 0.163 | 0.054 | 0.168 | 0.056 | 0.174 | 0.056 | 0.180 | 0.069 | 0.202 | 0.081 | 0.221 |
| ETTm2 | **0.111** | 0.250 | 0.159 | 0.313 | 0.112 | 0.251 | 0.112 | **0.248** | 0.117 | 0.256 | 0.171 | 0.321 | 0.119 | 0.262 | 0.130 | 0.271 |

and Swin4TS/CD exhibit impressive prediction performance, complementing each other to achieve the best.

However, on datasets like Traffic and Electricity that contain hundreds of channels, the performance of Swin4TS/CD is significantly inferior to Swin4TS/CI. This may be due to the complex correlations between these hundreds of channels, making it challenging for Swin4TS/CD to uncover the underlying correlations in limited data. Besides, recent studies (Han et al., 2023) have shown that due to the significant distribution shift between training and testing sets of existing datasets, the models with CI strategy would perform better than that with CD strategy especially on datasets with a mass of channels. Nevertheless, compared with other baselines with CD strategies, Swin4TS/CD still performs sufficiently well on all datasets. For example, on the 4 ETT datasets, Swin4TS/CD achieves nearly an average 10% improvement over the best of these models (ETTh1-5.14%, ETTh2-13.1%, ETTm1-6.2% and ETTm2-14.1%).

Univariate forecasting is additionally investigated on the scheme of Swin4TS. The performance on 4 ETT datasets is shown in Table 2 (full results are in Appendix B.2). It can be seen that Swin4TS achieves the SOTA on all 4 ETT datasets and outperforms all baselines. Especially on ETTh1, Swin4TS/CD surpasses the second-best result by 6.8% (0.069→0.074), and on ETTh2, Swin4TS/CI achieves 9.6% improvement (0.16→0.177).

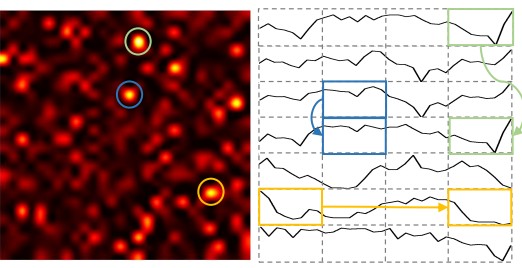

Figure 5: Attention map of Swin4TS/CD on ETTh1 dataset, showing the attention across both the time and channel dimensions.

**Correlations across channel and time** The above results show that Swin4TS/CD achieves the best performance on some datasets, such as ILI and ETTh1, and the MSE is sufficiently declined compared with Swin4TS/CI. This indicates that these improvements may be attributed to the correlations between channels discovered by Swin4TS/CD. In vision tasks, attention maps are often used to illustrate the distribution of attention or correlation on an image. Similarly, here we use the attention maps of Swin4TS/CD to visualize the attention distribution for the multivariate time series. The left side of Fig. 5 shows the attention map of the first window on ETTh1 dataset, and the corresponding time series are shown on the right side. The size of this window is $7 \times 4$, representing the number of patches in the channel and time dimensions, respectively. Thus it results in an attention map with size $28 \times 28$. We circle three representative bright spots on the map, corresponding to the locations of important attention, and match them with the actual positions in the time series. Clearly, the patches are not only correlated with patches in the same channel but also with patches from other channels. Intuitively, one can notice that two patches with similar trends seem to exhibit strong correlations. Nevertheless, this information could provide valuable guidance for understanding the complex patterns of multivariate time series.

**Ablation study** We further conduct ablation experiments on two key designs of Swin4TS. First is the shift window attention (referred as "shift" here), which is used to connect information between two consecutive windows. We remove this design by replacing it with normal window attention. Second is the hierarchical design (referred as "scale" here), which is used to explore temporal features at different scales. The ablation of it can be achieved by removing the downscaling operation after the first stage to ensure that all stages focus on the same scale. The results of Swin4TS/CD on

Table 3: Ablation study of Swin4TS/CD on ETTm1 and ETTm2. Bold/underline indicates the best/second in Swin4TS/CD.

| Models | | Swin4TS/CD | | | | | | | | Crossformer | | MICN | |
|---|---|---|---|---|---|---|---|---|---|---|---|---|---|
| | | shift&scale | | w/o shift | | w/o scale | | w/o shift&scale | | | | | |
| Metrics | | MSE | MAE | MSE | MAE | MSE | MAE | MSE | MAE | MSE | MAE | MSE | MAE |
| ETTm1 | 96 | 0.292 | 0.346 | **0.291** | **0.344** | 0.296 | 0.35 | 0.295 | 0.35 | 0.335 | 0.386 | 0.305 | 0.354 |
| | 192 | **0.334** | **0.369** | 0.335 | 0.37 | 0.336 | 0.374 | 0.339 | 0.376 | 0.383 | 0.429 | 0.353 | 0.390 |
| | 336 | **0.364** | **0.387** | 0.365 | 0.389 | 0.371 | 0.397 | 0.377 | 0.401 | 0.424 | 0.442 | 0.382 | 0.405 |
| | 720 | **0.402** | **0.417** | 0.427 | 0.423 | 0.423 | 0.432 | 0.434 | 0.434 | 0.598 | 0.545 | 0.445 | 0.442 |
| | Avg. | **0.348** | **0.380** | 0.355 | 0.382 | 0.357 | 0.388 | 0.361 | 0.390 | 0.435 | 0.450 | 0.371 | 0.398 |
| ETTm2 | 96 | **0.160** | **0.249** | 0.164 | 0.253 | 0.173 | 0.262 | 0.173 | 0.262 | 0.353 | 0.424 | 0.193 | 0.283 |
| | 192 | **0.219** | **0.290** | 0.224 | 0.293 | 0.228 | 0.298 | 0.23 | 0.3 | 0.531 | 0.513 | 0.248 | 0.321 |
| | 336 | **0.268** | **0.322** | 0.272 | 0.325 | 0.284 | 0.333 | 0.285 | 0.334 | 0.868 | 0.722 | 0.295 | 0.353 |
| | 720 | **0.344** | **0.371** | 0.351 | 0.376 | 0.361 | 0.381 | 0.363 | 0.382 | 1.880 | 1.010 | 0.427 | 0.447 |
| | Avg. | **0.248** | **0.308** | 0.253 | 0.312 | 0.262 | 0.319 | 0.263 | 0.320 | 0.908 | 0.667 | 0.291 | 0.351 |

two datasets are presented in Table 3, compared with two baselines in CD strategy (More ablation studies on Swin4TS/CI can be referred in Appendix C.2). It can be seen that after removing these two designs, the average prediction MSE of the model increases by 3.2% and 2.7% on ETTm1 and ETTm2, respectively. Removing either one of these two designs individually also significantly increases the prediction error. This indicates that these two key designs in Swin4TS play important roles in ensuring prediction accuracy.

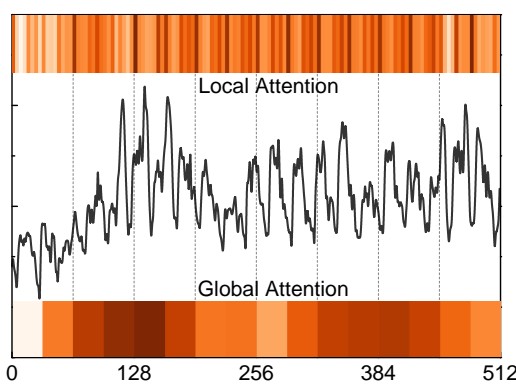

Figure 6: Attention map of Swin4TS/CI on ETTh1 dataset, showing the attention at local and global scale, respectively. The darker color means stronger attention.

**Hierarchical information** In the ablation experiment above, we confirmed the effectiveness of the hierarchical design. Here, we further explore the information extracted by Swin4TS at different scales. A 2-stage Swin4TS/CI framework is designed for the ETTh1 dataset, which enables us to examine the information at local and global scales. The information at different scales can be presented by the one-dimensional attention map (reduced by the attention matrix) of the corresponding stage. Fig. 6 shows a sample of time series of the ETTh1 dataset with local and global attention maps above and below it, respectively. It can be seen that the time series exhibit strong periodicity at the local scale, which is also reflected in the local attention map. At the global scale, there is a significant upward trend on the left, and this anomaly signal is also captured by global attention. Although the underlying patterns of time series represented by attention maps may be very complex, they qualitatively demonstrate the characteristics of time series at different scales, which helps improve the interpretability of predictions.

### 4.3 OTHER RESULTS

**Randomness test** Swin4TS demonstrates strong robustness to different random seeds or model initialization. See Appendix C.1.

**Effect of channel order** A shuffled initial channel order for Swin4TS/CD benefits the performance. See Appendix C.3

**Varying hierarchical design** Different hierarchical designs capture different scale information hidden in the time series which vary much among different datasets. See Appendix C.4

**Varying historical series length** In most cases, longer historical series length leads to better prediction. See Appendix C.5

**Effect of dynamic covariate** Dynamic covariate facilitates the prediction for datasets with obvious periodicity. See Appendix C.6

**Transferability of Swin4TS/CI** The model trained on data from one channel can still perform well in predicting unseen data from other channels. See Appendix C.7.

## 5 COMPUTATIONAL COMPLEXITY ANALYSIS

For the historical series with $M$ channels of length $L$, the complexity of Swin4TS/CI is $O(ML)$ as aforementioned. Despite considering both channel and time, the complexity of Swin4TS/CD is still linear with the number of windows as the complexity within a window is fixed by pre-defined window size. Thus one can easily prove that Swin4TS/CD shares the same complexity $O(ML)$ with Swin4TS/CI.

Table 4 shows the complexity for Swin4TS and other Transformer-based models. Although the earlier models (Transformer, Informer, Autoformer and FEDformer) adopt the CD strategy, they do not consider the correlations between channels explicitly and just fuse multiple channels by an embedding layer, and thus the complexity of these models is irrelevant with channels. The recent models, PatchTST and Crossformer, show the relevance with channels. But the complexity of these models is essentially quadratic with $L$. To the best of our knowledge, Swin4TS is the first Transformer-based model owning linear complexity with both $L$ and $M$. Besides, Table 4 also shows the inference efficiency on the Electricity dataset. As can be seen, Swin4TS achieves a good balance between inference time and memory usage. Especially when dealing with large datasets, the inference efficiency of Swin4TS/CI is significantly better than all Transformer-based algorithms.

Table 4: Computational complexity analysis and inference efficiency on Electricity for Swin4TS and other Transformer-based models.

| Method | Encoder layer | Decoder layer | Time (ms) | Memory (GB) |
|---|---|---|---|---|
| Transformer (Vaswani et al., 2017) | $O(L^2)$ | $O(T(T+L))$ | 21.1 | 5.8 |
| Informer (Zhou et al., 2021) | $O(L\log L)$ | $O(T(T+\log L))$ | 43.8 | 3.9 |
| Autoformer (Wu et al., 2021) | $O(L\log L)$ | $O((\frac{L}{2}+T)\log(\frac{L}{2}+T))$ | 123.4 | 7.6 |
| FEDformer (Zhou et al., 2022) | $O(L)$ | $O(\frac{L}{2}+T)$ | 33.5 | 4.3 |
| PatchTST (Nie et al., 2023) | $O(ML^2)$ | $O(MT)$ | 15.4 | 3.2 |
| Crossformer (Zhang & Yan, 2023) | $O(ML^2)$ | $O(MT(T+L))$ | 25.2 | **2.0** |
| Swin4TS/CI (**ours**) | $O(ML)$ | $O(MT)$ | **11.3** | **2.0** |
| Swin4TS/CD (**ours**) | $O(ML)$ | $O(MT)$ | 45.3 | 5.4 |

## 6 CONCLUSION AND DISSUSION

In this work, we have proposed the Swin4TS algorithm for long-term time series forecasting. Swin4TS incorporates two key designs, window-based attention and hierarchical representation, from the Swin Transformer to model the time series. It can easily adapt to both channel-dependent and channel-independent strategies, resulting in two variants: Swin4TS/CD and Swin4TS/CI. Swin4TS/CD can simultaneously capture correlations in both channel and time dimensions, while Swin4TS/CI considers the channels independently and thus shows impressive computational efficiency. These two variants complement each other and outperform the latest baselines, achieving state-of-the-art performance on 8 benchmark datasets.

The effectiveness of Swin4TS confirms that time series and image (partitioned into patch series) can be modeled using the same framework, thanks to their similarity in data structure. Although images have inherent inductive biases such as shift invariance, and time series also have inherent characteristics such as trend and seasonality, these features can be integrated into underlying patterns and learned by Transformer. To further confirm this point, we designed the TNT4TS architecture. TNT (Transformer in Transformer (Han et al., 2021)) is a classic ViT architecture that can effectively capture both local and global information. When applied to time series forecasting, we found that it also performs quite well (see the Appendix E). Both Swin4TS and TNT4TS differ from the original ViT models used for processing images, only in that time series are one-dimensional, while images are two-dimensional. We hope that this work can inspire more similar endeavors to utilize advanced ViT models for other time series tasks, e.g., anomaly detection, classification, and imputation.

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

## A  EXPERIMENTAL DETAILS

### A.1  DESCRIPTION OF DATASET

We use 8 widely-used datasets in the main text. The details are listed in the following:

- Traffic[2] describes the hourly 862 road occupancy rates of San Francisco freeways from 2015 to 2016.
- Electricity[3] measures the electricity consumption of 321 clients from 2012 to 2014.
- Weather[4] that contains 21 climate features, such as temperature, pressure and humidity, in 10-minute-level for 2020 whole year in Germany.
- ILI[5] describes the weekly ratio of patients seen with influenza-like illness and the number of patients. It includes data from the Centers for Disease Control and Prevention of the United States from 2002 to 2021.
- ETT[6] (Electricity Transformer Temperature) measures 6 power load features and 1 oil temperature feature, which consists of two hourly-level datasets (ETTh1, ETTh2) and two 15-minute-level datasets (ETTm1 and ETTm2), from July 2016 to July 2018.

The statistics of these datasets are listed in Table 5.

Table 5: The statistics of the 8 benchmark datasets.

| Datasets | Weather | Traffic | Electricity | ILI | ETTh1 | ETTh2 | ETTm1 | ETTm2 |
|---|---|---|---|---|---|---|---|---|
| Number of series | 21 | 862 | 321 | 7 | 7 | 7 | 7 | 7 |
| Timesteps | 52696 | 17544 | 26304 | 966 | 17420 | 17420 | 69680 | 69680 |

### A.2  REPRODUCTION DETAILS FOR SWIN4TS

The input series of Swin4TS are firstly normalized by the mean and standard value, which further are used to rescale the output. For the 4 ETT datasets, each one is split in chronological order with 60% for training, 20% for validation, and 20% for testing. While for the left 4 datasets, each one is split in chronological order with 70% for training, 10% for validation, and 20% for testing. The used 8 benchmark datasets can be categorized into two sets: Weather, Traffic and Electricity are regarded as large datasets, and the left containing 7 channels are regarded as small datasets. In all our experiments, the maximum training epoch is set to 100, and the training process will stop early if the validation loss does not decrease within 20 epochs. Optimizer uses ADAM for L2 loss with an initial learning rate of $5 \times 10^{-4}$. Batch size is set to 64 and 128 for large datasets and small datasets, respectively. All the experiments are implemented in Pytorch 1.12 and conducted on NVIDIA A100-SXM4-40GB GPU.

Table 6 shows the details of hierarchical architecture of Swin4TS/CI. Window size, heads and layers in each stage are uniformly set to 8 (except 9 for ILI), 4 (except 16 for ETTm1 and ETTm2) and 2, respectively. The small datasets adopt a 2-stage hierarchical design, except that the ILI dataset adopts only 1 stage architecture. The large datasets adopt a 4-stage hierarchical design, except that the Weather dataset adopts a 2-stage architecture.

Table 7 shows the details of hierarchical architecture of Swin4TS/CD. As same as the choice of Swin4TS/CI, Weather and 4 ETT datasets adopt a 2-stage hierarchical design, and ILI dataset adopts only 1-stage architecture. However, due to the limitation of memory usage, Traffic and Electricity here also take a 2-stage architecture.

---

[2]https://pems.dot.ca.gov/
[3]https://archive.ics.uci.edu/ml/datasets/
[4]https://www.bgc-jena.mpg.de/wetter/
[5]https://gis.cdc.gov/grasp/fluview/fluportaldashboard.html
[6]https://github.com/zhouhaoyi/ETDataset

Table 6: Key parameters of the hierarchical architecture for Swin4TS/CI.

| Parameters | | Weather | Traffic | Electricity | ILI | ETTh1 | ETTh2 | ETTm1 | ETTm2 |
|---|---|---|---|---|---|---|---|---|---|
| window_size | | 8 | 8 | 8 | 9 | 8 | 8 | 8 | 8 |
| stage 1 | downscaling | 8 | 8 | 8 | 2 | 4 | 4 | 4 | 4 |
| | heads | 4 | 4 | 4 | 4 | 4 | 4 | 16 | 16 |
| | layers | 2 | 2 | 2 | 2 | 2 | 2 | 2 | 2 |
| stage 2 | downscaling | 8 | 2 | 2 | - | 8 | 8 | 8 | 8 |
| | heads | 4 | 4 | 4 | - | 4 | 4 | 16 | 16 |
| | layers | 2 | 2 | 2 | - | 2 | 2 | 2 | 2 |
| stage 3 | downscaling | - | 2 | 2 | - | - | - | - | - |
| | heads | - | 4 | 4 | - | - | - | - | - |
| | layers | - | 2 | 2 | - | - | - | - | - |
| stage 4 | downscaling | - | 2 | 2 | - | - | - | - | - |
| | heads | - | 4 | 4 | - | - | - | - | - |
| | layers | - | 2 | 2 | - | - | - | - | - |

Table 7: Key parameters of the hierarchical architecture for Swin4TS/CD.

| Parameters | | Weather | Traffic | Electricity | ILI | ETTh1 | ETTh2 | ETTm1 | ETTm2 |
|---|---|---|---|---|---|---|---|---|---|
| window_size_t | | 8 | 8 | 4 | 9 | 8 | 16 | 8 | 4 |
| window_size_c | | 7 | 11 | 4 | 7 | 7 | 7 | 7 | 7 |
| stage 1 | downscaling_t | 8 | 8 | 4 | 3 | 8 | 8 | 4 | 4 |
| | downscaling_c | 1 | 13 | 4 | 1 | 1 | 1 | 1 | 1 |
| | heads | 4 | 4 | 4 | 16 | 16 | 4 | 4 | 8 |
| | layers | 2 | 2 | 2 | 2 | 2 | 2 | 2 | 2 |
| stage 2 | downscaling_t | 8 | 8 | 4 | - | 4 | 4 | 8 | 8 |
| | downscaling_c | 1 | 1 | 2 | - | 1 | 1 | 1 | 1 |
| | heads | 4 | 4 | 4 | - | 16 | 4 | 4 | 4 |
| | layers | 2 | 2 | 2 | - | 2 | 2 | 2 | 2 |

### A.3 REPRODUCTION DETAILS FOR BASELINES

In this work, we have introduced a total of 7 the most recent and popular baseline models used for long-term forecasting: PatchTST, DLinear, MICN, TimesNet, N-HiTS, Crossformer, FEDformer, and Autoformer. For all these models, we use the open-source codes from their respective GitHub repositories, and adopt their default hyperparameters to train the models. The historical series length $L$ in the original paper for MICN, TimesNet, Crossformer, FEDformer, and Autoformer is set as 96, and it for DLinear is set as 336, while it for PatchTST and our Swin4TS is set as 512. To make a fair comparison, we evaluate the involved baseline models on $L = 96$, 336, and 512 for each task, respectively, and then select the best one as their final result. This ensures that the results presented here will not be worse than those presented in their original papers. It is important to note that for Swin4TS and all baseline models being compared, we set the batch size to 1 during testing. This is because if the batch size is not back to 1 during testing, it may result in the omission of some test samples from the last batch and lead to inaccurate testing. Thus the results of PatchTST shown in the present work are slightly different from that in its original paper.

In addition, we also notice some excellent recent works such as MLP-based TSMixer (Ekambaram et al., 2023) and TiDE (Das et al., 2023). They focus more on the efficiency of model training and inference, but their performance is basically consistent with PatchTST. There are also some algorithms based on contrastive learning, such as TS2Vec (Yue et al., 2021), CoST (Woo et al., 2023), LaST (Wang et al., 2022c), and so on. These works emphasize generalizability on time series analysis tasks including time series prediction, anomaly detection, classification, imputation, etc. Therefore, due to page limitations, we do not compare with these algorithms.

## B FULL RESULTS

### B.1 FULL RESULTS OF MULTIVARIATE FORECASTING

The full results of multivariate forecasting for Swin4TS/CI and Swin4TS/CD compared with other baselines are shown in Table 8. These algorithms are evaluated on 8 datasets with a total of 32 pre-

Table 8: Full results of multivariate long-term series forecasting on 8 datasets. Bold/underline indicates the best/second. Algorithm with * suggests the use of CI strategy otherwise the CD strategy. Prediction length $T \in \{24, 36, 48, 60\}$ for ILI dataset and $T \in \{96, 192, 336, 720\}$ for the others.

| Models | | Swin4TS/CI* (ours) | | Swin4TS/CD* (ours) | | PatchTST/64* (2023) | | DLinear* (2023) | | MICN (2023) | | N-HiTS (2023) | | TimesNet (2023) | | Crossformer (2023) | | FEDformer (2022) | | Autoformer 2021 | |
|---|---|---|---|---|---|---|---|---|---|---|---|---|---|---|---|---|---|---|---|---|---|
| Metrics | | MSE | MAE | MSE | MAE | MSE | MAE | MSE | MAE | MSE | MAE | MSE | MAE | MSE | MAE | MSE | MAE | MSE | MAE | MSE | MAE |
| Weather | 96 | **0.143** | **0.191** | 0.147 | 0.194 | 0.148 | 0.197 | 0.176 | 0.237 | 0.181 | 0.249 | 0.158 | 0.195 | 0.150 | 0.219 | 0.170 | 0.228 | 0.238 | 0.314 | 0.249 | 0.329 |
| | 192 | **0.189** | **0.235** | 0.191 | 0.238 | 0.191 | 0.239 | 0.220 | 0.282 | 0.219 | 0.276 | 0.211 | 0.247 | 0.194 | 0.262 | 0.215 | 0.263 | 0.275 | 0.329 | 0.325 | 0.370 |
| | 336 | **0.238** | 0.278 | 0.242 | **0.277** | 0.243 | 0.280 | 0.265 | 0.319 | 0.259 | 0.310 | 0.274 | 0.300 | 0.243 | 0.304 | 0.272 | 0.301 | 0.339 | 0.377 | 0.351 | 0.391 |
| | 720 | **0.312** | **0.328** | 0.315 | 0.330 | 0.312 | 0.331 | 0.323 | 0.362 | 0.307 | 0.343 | 0.351 | 0.353 | 0.315 | 0.359 | 0.342 | 0.352 | 0.389 | 0.409 | 0.415 | 0.426 |
| Traffic | 96 | **0.329** | **0.241** | 0.512 | 0.325 | 0.360 | 0.248 | 0.410 | 0.282 | 0.457 | 0.295 | 0.402 | 0.282 | 0.591 | 0.318 | 0.496 | 0.279 | 0.576 | 0.359 | 0.597 | 0.371 |
| | 192 | **0.351** | **0.252** | 0.522 | 0.333 | 0.379 | 0.256 | 0.423 | 0.287 | 0.468 | 0.303 | 0.420 | 0.297 | 0.620 | 0.342 | 0.499 | 0.276 | 0.610 | 0.380 | 0.607 | 0.382 |
| | 336 | **0.367** | **0.259** | 0.529 | 0.341 | 0.392 | 0.265 | 0.436 | 0.296 | 0.480 | 0.301 | 0.448 | 0.313 | 0.623 | 0.339 | 0.534 | 0.298 | 0.608 | 0.375 | 0.623 | 0.387 |
| | 720 | **0.377** | **0.266** | 0.541 | 0.343 | 0.453 | 0.312 | 0.466 | 0.315 | 0.529 | 0.323 | 0.539 | 0.353 | 0.648 | 0.344 | 0.564 | 0.308 | 0.621 | 0.375 | 0.639 | 0.395 |
| Electricity | 96 | **0.127** | **0.221** | 0.159 | 0.271 | 0.129 | 0.223 | 0.140 | 0.237 | 0.156 | 0.265 | 0.147 | 0.249 | 0.168 | 0.272 | 0.149 | 0.249 | 0.186 | 0.302 | 0.196 | 0.313 |
| | 192 | **0.144** | **0.237** | 0.170 | 0.268 | 0.148 | 0.241 | 0.153 | 0.249 | 0.165 | 0.275 | 0.167 | 0.269 | 0.184 | 0.289 | 0.164 | 0.262 | 0.197 | 0.311 | 0.211 | 0.324 |
| | 336 | **0.16** | **0.255** | 0.175 | 0.282 | 0.165 | 0.261 | 0.169 | 0.267 | 0.180 | 0.290 | 0.186 | 0.290 | 0.198 | 0.300 | 0.196 | 0.295 | 0.213 | 0.328 | 0.214 | 0.327 |
| | 720 | **0.197** | **0.288** | 0.201 | 0.299 | 0.198 | 0.290 | 0.203 | 0.301 | 0.210 | 0.318 | 0.243 | 0.340 | 0.220 | 0.320 | 0.266 | 0.356 | 0.233 | 0.344 | 0.236 | 0.342 |
| ILI | 96 | **1.740** | **0.854** | 1.826 | 0.909 | 2.044 | 0.876 | 2.215 | 1.081 | 2.453 | 1.081 | 1.862 | 0.869 | 2.317 | 0.934 | 3.064 | 1.180 | 2.624 | 1.095 | 2.906 | 1.182 |
| | 192 | **1.713** | **0.859** | 1.729 | 0.878 | 2.012 | 0.938 | 1.963 | 0.963 | 2.356 | 1.053 | 2.071 | 0.934 | 1.972 | 0.920 | 3.150 | 1.188 | 2.516 | 1.021 | 2.585 | 1.038 |
| | 336 | 1.718 | 0.877 | **1.539** | **0.797** | 1.768 | 0.897 | 2.130 | 1.024 | 2.749 | 1.145 | 2.134 | 0.932 | 2.238 | 0.940 | 3.332 | 1.227 | 2.505 | 1.041 | 3.024 | 1.145 |
| | 720 | 1.790 | 0.908 | **1.533** | **0.813** | 2.043 | 0.971 | 2.368 | 1.096 | 2.636 | 1.141 | 2.137 | 0.968 | 2.027 | 0.928 | 3.740 | 1.294 | 2.742 | 1.122 | 2.761 | 1.114 |
| ETTh1 | 96 | 0.366 | 0.394 | **0.365** | **0.392** | 0.377 | 0.405 | 0.375 | 0.399 | 0.431 | 0.442 | 0.378 | 0.393 | 0.384 | 0.402 | 0.451 | 0.461 | 0.376 | 0.415 | 0.435 | 0.446 |
| | 192 | 0.403 | 0.420 | **0.400** | **0.414** | 0.411 | 0.428 | 0.405 | 0.416 | 0.443 | 0.461 | 0.427 | 0.436 | 0.436 | 0.429 | 0.422 | 0.447 | 0.423 | 0.446 | 0.456 | 0.457 |
| | 336 | **0.425** | **0.433** | 0.425 | 0.440 | 0.432 | 0.445 | 0.439 | 0.443 | 0.502 | 0.484 | 0.458 | 0.484 | 0.491 | 0.469 | 0.437 | 0.462 | 0.444 | 0.462 | 0.486 | 0.487 |
| | 720 | 0.448 | 0.463 | **0.432** | **0.456** | 0.456 | 0.473 | 0.472 | 0.490 | 0.622 | 0.596 | 0.472 | 0.561 | 0.521 | 0.500 | 0.505 | 0.513 | 0.469 | 0.492 | 0.515 | 0.517 |
| ETTh2 | 96 | 0.272 | 0.334 | **0.264** | **0.330** | 0.275 | 0.339 | 0.289 | 0.353 | 0.296 | 0.362 | 0.274 | 0.345 | 0.383 | 0.420 | 0.894 | 0.671 | 0.332 | 0.374 | 0.332 | 0.368 |
| | 192 | 0.336 | 0.377 | **0.331** | **0.375** | 0.339 | 0.380 | 0.383 | 0.418 | 0.406 | 0.427 | 0.353 | 0.401 | 0.409 | 0.436 | 0.886 | 0.666 | 0.407 | 0.446 | 0.426 | 0.434 |
| | 336 | 0.362 | 0.404 | **0.358** | **0.401** | 0.365 | 0.404 | 0.448 | 0.465 | 0.513 | 0.498 | 0.382 | 0.425 | 0.389 | 0.435 | 1.115 | 0.764 | 0.400 | 0.447 | 0.477 | 0.479 |
| | 720 | **0.384** | **0.427** | 0.386 | 0.427 | 0.390 | 0.430 | 0.605 | 0.551 | 0.875 | 0.660 | 0.625 | 0.557 | 0.460 | 0.476 | 1.197 | 0.815 | 0.412 | 0.469 | 0.453 | 0.490 |
| ETTm1 | 96 | **0.283** | **0.341** | 0.292 | 0.346 | 0.292 | 0.346 | 0.299 | 0.343 | 0.305 | 0.354 | 0.302 | 0.35 | 0.338 | 0.375 | 0.335 | 0.386 | 0.326 | 0.390 | 0.510 | 0.492 |
| | 192 | **0.325** | **0.366** | 0.334 | 0.369 | 0.331 | 0.370 | 0.335 | 0.365 | 0.353 | 0.390 | 0.347 | 0.383 | 0.374 | 0.387 | 0.383 | 0.429 | 0.365 | 0.415 | 0.514 | 0.495 |
| | 336 | **0.355** | **0.383** | 0.364 | 0.387 | 0.367 | 0.391 | 0.369 | 0.386 | 0.382 | 0.405 | 0.369 | 0.402 | 0.410 | 0.411 | 0.424 | 0.442 | 0.392 | 0.425 | 0.510 | 0.492 |
| | 720 | **0.401** | **0.413** | 0.402 | 0.417 | 0.421 | 0.420 | 0.425 | 0.421 | 0.445 | 0.442 | 0.431 | 0.441 | 0.478 | 0.450 | 0.598 | 0.545 | 0.446 | 0.458 | 0.527 | 0.493 |
| ETTm2 | 96 | 0.163 | 0.251 | **0.160** | **0.249** | 0.166 | 0.256 | 0.167 | 0.260 | 0.193 | 0.283 | 0.176 | 0.255 | 0.184 | 0.272 | 0.353 | 0.424 | 0.180 | 0.271 | 0.205 | 0.293 |
| | 192 | **0.216** | **0.292** | 0.219 | 0.290 | 0.221 | 0.294 | 0.224 | 0.303 | 0.248 | 0.321 | 0.245 | 0.305 | 0.240 | 0.309 | 0.531 | 0.513 | 0.252 | 0.318 | 0.278 | 0.336 |
| | 336 | **0.268** | **0.323** | 0.268 | 0.322 | 0.271 | 0.327 | 0.281 | 0.342 | 0.295 | 0.353 | 0.295 | 0.346 | 0.305 | 0.349 | 0.868 | 0.722 | 0.324 | 0.364 | 0.343 | 0.379 |
| | 720 | 0.354 | 0.378 | **0.344** | **0.371** | 0.361 | 0.384 | 0.397 | 0.421 | 0.427 | 0.447 | 0.401 | 0.413 | 0.400 | 0.407 | 1.880 | 1.010 | 0.410 | 0.420 | 0.414 | 0.419 |

diction tasks. It can be seen that overall, Swin4TS surpasses the previous state-of-the-art algorithms in nearly all 32 tasks, demonstrating excellent prediction performance. Among them, Swin4TS/CI performs better than the previous best results in almost all tasks, but its performance is slightly worse than Swin4TS/CD in the ILI and 4 ETT datasets. Although the performance of Swin4TS/CD on three large datasets does not reach the best, it shows significant improvement compared with other algorithms in the CD category (namely MICN, Crossformer, and TimesNet, FEDformer and Autoformer). For example, comparing with the best of these models, Swin4TS/CD achieves nearly an average 10% improvement on the 4 ETT datasets (ETTh1-5.14%, ETTh2-13.1%, ETTm1-6.2% and ETTm2-14.1%).

## B.2 Full results of univariate forecasting

We also conduct the experiment of univariate long-term forecasting on 4 ETT datasets. There is a feature "oil temperature" within those datasets, which is the target univariate series to forecast. For Swin4TS/CD, full channels are utilized to predict the target channel. While for Swin4TS/CI, only the target channel is involved in the model. The full results containing 6 baseline methods are shown in Table 9. The results of MICN and TimesNet are obtained by running their source code, while the remaining baseline results are referred to (Nie et al., 2023). It can be seen that Swin4TS achieves the SOTA on all 4 ETT datasets. Especially on ETTh1, Swin4TS/CD surpasses the second-best result by 6.8% (0.069→0.074). And on ETTh2, Swin4TS/CI surpasses the second-best result by 9.6% (0.16→0.177).

## B.3 Full results of baselines with different length of historical series

The full results of 3 newest baselines (MICN, TimesNet and Crossformer) with different lengths of $L$ are shown in Table 10. Here 3 input lengths $L = 96$, 336 and 512 are considered. As can be seen, there is no single $L$ for any baseline that consistently achieves the best performance across all datasets. For example, the best performance of MICN on large datasets occurs at $L = 512$, while on small datasets, the optimal $L$ varies. Similar conclusions can be drawn for Crossformer. However, as shown in Table 13, longer inputs for Swin4TS tend to yield consistently better results in most cases.

Table 9: Univariate long-term forecasting results with Swin4TS. ETT datasets are used with historical series length $L = 512$ and prediction lengths $T \in \{96, 192, 336, 720\}$. Bold/underline indicates the best/second. Algorithm with * suggests the use of CI strategy otherwise the CD strategy.

| Models | | Swin4TS/CI* (ours) | | Swin4TS/CD (ours) | | PatchTST/64* (2023) | | DLinear* (2023) | | MICN (2023) | | TimesNet (2023) | | FEDformer (2022) | | Autoformer (2021) | |
|---|---|---|---|---|---|---|---|---|---|---|---|---|---|---|---|---|---|
| Metrics | | MSE | MAE | MSE | MAE | MSE | MAE | MSE | MAE | MSE | MAE | MSE | MAE | MSE | MAE | MSE | MAE |
| ETTh1 | 96 | 0.057 | 0.187 | 0.067 | 0.206 | 0.058 | 0.187 | 0.056 | 0.180 | 0.072 | 0.218 | 0.070 | 0.205 | 0.079 | 0.215 | 0.071 | 0.206 |
| | 192 | 0.073 | 0.215 | 0.068 | 0.208 | 0.073 | 0.214 | 0.071 | 0.204 | 0.080 | 0.220 | 0.085 | 0.229 | 0.104 | 0.245 | 0.114 | 0.262 |
| | 336 | 0.079 | 0.226 | 0.069 | 0.212 | 0.079 | 0.225 | 0.098 | 0.244 | 0.139 | 0.298 | 0.095 | 0.245 | 0.119 | 0.270 | 0.107 | 0.258 |
| | 720 | 0.083 | 0.230 | 0.073 | 0.220 | 0.086 | 0.234 | 0.189 | 0.359 | 0.195 | 0.359 | 0.093 | 0.244 | 0.142 | 0.299 | 0.126 | 0.283 |
| | Avg. | 0.073 | 0.214 | 0.069 | 0.211 | 0.074 | 0.215 | 0.104 | 0.247 | 0.122 | 0.274 | 0.086 | 0.231 | 0.111 | 0.257 | 0.105 | 0.252 |
| ETTh2 | 96 | 0.126 | 0.280 | 0.176 | 0.345 | 0.132 | 0.285 | 0.131 | 0.279 | 0.134 | 0.286 | 0.170 | 0.325 | 0.128 | 0.271 | 0.153 | 0.306 |
| | 192 | 0.158 | 0.317 | 0.174 | 0.344 | 0.171 | 0.329 | 0.176 | 0.329 | 0.174 | 0.334 | 0.191 | 0.343 | 0.185 | 0.330 | 0.204 | 0.351 |
| | 336 | 0.165 | 0.327 | 0.175 | 0.343 | 0.185 | 0.347 | 0.209 | 0.367 | 0.199 | 0.359 | 0.204 | 0.361 | 0.231 | 0.378 | 0.246 | 0.389 |
| | 720 | 0.192 | 0.354 | 0.174 | 0.343 | 0.220 | 0.377 | 0.276 | 0.426 | 0.322 | 0.462 | 0.219 | 0.385 | 0.278 | 0.420 | 0.268 | 0.409 |
| | Avg. | 0.160 | 0.319 | 0.175 | 0.344 | 0.177 | 0.335 | 0.198 | 0.350 | 0.207 | 0.360 | 0.196 | 0.353 | 0.206 | 0.350 | 0.218 | 0.364 |
| ETTm1 | 96 | 0.026 | 0.122 | 0.035 | 0.149 | 0.026 | 0.122 | 0.028 | 0.123 | 0.028 | 0.125 | 0.029 | 0.132 | 0.033 | 0.140 | 0.056 | 0.183 |
| | 192 | 0.039 | 0.150 | 0.046 | 0.167 | 0.040 | 0.152 | 0.045 | 0.156 | 0.041 | 0.153 | 0.049 | 0.170 | 0.058 | 0.186 | 0.081 | 0.216 |
| | 336 | 0.053 | 0.175 | 0.058 | 0.186 | 0.053 | 0.174 | 0.061 | 0.182 | 0.063 | 0.187 | 0.064 | 0.193 | 0.084 | 0.231 | 0.076 | 0.218 |
| | 720 | 0.069 | 0.202 | 0.079 | 0.222 | 0.073 | 0.205 | 0.08 | 0.21 | 0.092 | 0.232 | 0.083 | 0.223 | 0.102 | 0.250 | 0.110 | 0.267 |
| | Avg. | 0.047 | 0.162 | 0.055 | 0.181 | 0.048 | 0.163 | 0.054 | 0.168 | 0.056 | 0.174 | 0.056 | 0.180 | 0.069 | 0.202 | 0.081 | 0.221 |
| ETTm2 | 96 | 0.063 | 0.185 | 0.126 | 0.281 | 0.065 | 0.188 | 0.063 | 0.183 | 0.065 | 0.187 | 0.103 | 0.247 | 0.067 | 0.198 | 0.065 | 0.189 |
| | 192 | 0.093 | 0.231 | 0.144 | 0.295 | 0.093 | 0.232 | 0.092 | 0.227 | 0.094 | 0.232 | 0.162 | 0.312 | 0.102 | 0.245 | 0.118 | 0.256 |
| | 336 | 0.121 | 0.267 | 0.162 | 0.316 | 0.119 | 0.265 | 0.119 | 0.261 | 0.130 | 0.275 | 0.213 | 0.362 | 0.130 | 0.279 | 0.154 | 0.305 |
| | 720 | 0.167 | 0.319 | 0.206 | 0.360 | 0.170 | 0.321 | 0.175 | 0.32 | 0.180 | 0.328 | 0.208 | 0.365 | 0.178 | 0.325 | 0.182 | 0.335 |
| | Avg. | 0.111 | 0.250 | 0.159 | 0.313 | 0.112 | 0.251 | 0.112 | 0.248 | 0.117 | 0.256 | 0.171 | 0.321 | 0.119 | 0.262 | 0.130 | 0.271 |

Table 10: Full results of the 3 newest baselines, MICN, TimesNet and Crossformer, with 3 historical series lengths $L = 96, 336$ and $512$ are considered. The results of different $L$ of each baseline are compared internally. The bold indicates the best. Prediction length $T \in \{24, 36, 48, 60\}$ for ILI dataset and $T \in \{96, 192, 336, 720\}$ for the others.

| Models | | MICN (96) | | MICN (336) | | MICN (512) | | Timesnet (96) | | Timesnet (336) | | Timesnet (512) | | Crossformer (96) | | Crossformer (336) | | Crossformer (512) | |
|---|---|---|---|---|---|---|---|---|---|---|---|---|---|---|---|---|---|---|---|
| Metrics | | MSE | MAE | MSE | MAE | MSE | MAE | MSE | MAE | MSE | MAE | MSE | MAE | MSE | MAE | MSE | MAE | MSE | MAE |
| Weather | 96 | 0.161 | 0.229 | 0.170 | 0.235 | 0.181 | 0.249 | 0.172 | 0.220 | 0.170 | 0.228 | 0.159 | 0.214 | 0.157 | 0.228 | 0.161 | 0.240 | 0.150 | 0.219 |
| | 192 | 0.220 | 0.281 | 0.218 | 0.279 | 0.219 | 0.276 | 0.219 | 0.261 | 0.215 | 0.263 | 0.222 | 0.270 | 0.207 | 0.276 | 0.193 | 0.262 | 0.194 | 0.262 |
| | 336 | 0.278 | 0.331 | 0.281 | 0.329 | 0.259 | 0.310 | 0.280 | 0.306 | 0.272 | 0.301 | 0.279 | 0.310 | 0.261 | 0.320 | 0.247 | 0.308 | 0.243 | 0.304 |
| | 720 | 0.311 | 0.356 | 0.327 | 0.366 | 0.307 | 0.343 | 0.365 | 0.359 | 0.342 | 0.352 | 0.343 | 0.355 | 0.364 | 0.391 | 0.312 | 0.356 | 0.315 | 0.359 |
| Traffic | 96 | 0.519 | 0.309 | 0.488 | 0.298 | 0.457 | 0.295 | 0.593 | 0.321 | 0.591 | 0.318 | 0.600 | 0.321 | 0.534 | 0.301 | 0.499 | 0.271 | 0.496 | 0.279 |
| | 192 | 0.537 | 0.315 | 0.481 | 0.304 | 0.468 | 0.303 | 0.617 | 0.336 | 0.620 | 0.342 | 0.614 | 0.329 | 0.558 | 0.314 | 0.494 | 0.403 | 0.499 | 0.276 |
| | 336 | 0.534 | 0.313 | 0.489 | 0.296 | 0.480 | 0.301 | 0.629 | 0.336 | 0.623 | 0.339 | 0.659 | 0.349 | 0.530 | 0.300 | 0.551 | 0.306 | 0.534 | 0.298 |
| | 720 | 0.577 | 0.325 | 0.516 | 0.313 | 0.529 | 0.323 | 0.640 | 0.350 | 0.648 | 0.344 | 0.654 | 0.347 | 0.573 | 0.313 | 0.579 | 0.310 | 0.564 | 0.308 |
| Electricity | 96 | 0.164 | 0.269 | 0.155 | 0.265 | 0.156 | 0.265 | 0.168 | 0.272 | 0.177 | 0.284 | 0.181 | 0.286 | 0.149 | 0.249 | 0.143 | 0.244 | 0.153 | 0.257 |
| | 192 | 0.177 | 0.285 | 0.180 | 0.288 | 0.165 | 0.275 | 0.184 | 0.289 | 0.183 | 0.286 | 0.194 | 0.295 | 0.164 | 0.262 | 0.167 | 0.271 | 0.218 | 0.314 |
| | 336 | 0.193 | 0.304 | 0.201 | 0.307 | 0.180 | 0.290 | 0.198 | 0.300 | 0.204 | 0.305 | 0.200 | 0.302 | 0.196 | 0.295 | 0.195 | 0.299 | 0.223 | 0.315 |
| | 720 | 0.212 | 0.321 | 0.280 | 0.366 | 0.210 | 0.318 | 0.220 | 0.320 | 0.216 | 0.318 | 0.227 | 0.324 | 0.266 | 0.356 | 0.268 | 0.355 | 0.264 | 0.353 |
| ILI | 24 | 2.684 | 1.112 | 2.453 | 1.081 | 2.442 | 1.071 | 2.317 | 0.934 | 2.220 | 0.978 | 2.220 | 0.993 | 3.041 | 1.186 | 3.078 | 1.202 | 3.064 | 1.180 |
| | 36 | 2.667 | 1.068 | 2.356 | 1.053 | 2.737 | 1.140 | 1.972 | 0.920 | 2.318 | 1.031 | 2.498 | 1.072 | 3.406 | 1.232 | 3.206 | 1.154 | 3.150 | 1.188 |
| | 48 | 2.558 | 1.052 | 2.749 | 1.145 | 2.393 | 1.076 | 2.238 | 0.940 | 2.122 | 1.006 | 2.540 | 1.080 | 3.459 | 1.221 | 3.505 | 1.252 | 3.332 | 1.227 |
| | 60 | 2.747 | 1.110 | 2.636 | 1.141 | 2.831 | 1.186 | 2.027 | 0.928 | 1.975 | 0.975 | 2.091 | 0.990 | 3.640 | 1.305 | 3.740 | 1.294 | 3.745 | 1.314 |
| ETTh1 | 96 | 0.431 | 0.442 | 0.423 | 0.444 | 0.406 | 0.431 | 0.384 | 0.402 | 0.421 | 0.436 | 0.442 | 0.457 | 0.410 | 0.432 | 0.457 | 0.468 | 0.451 | 0.461 |
| | 192 | 0.443 | 0.461 | 0.449 | 0.462 | 0.497 | 0.502 | 0.436 | 0.429 | 0.474 | 0.477 | 0.491 | 0.491 | 0.469 | 0.470 | 0.426 | 0.441 | 0.422 | 0.447 |
| | 336 | 0.502 | 0.502 | 0.577 | 0.548 | 0.514 | 0.524 | 0.491 | 0.469 | 0.487 | 0.477 | 0.489 | 0.491 | 0.440 | 0.461 | 0.447 | 0.460 | 0.437 | 0.462 |
| | 720 | 0.622 | 0.596 | 0.701 | 0.640 | 0.693 | 0.643 | 0.521 | 0.500 | 0.538 | 0.518 | 0.555 | 0.528 | 0.519 | 0.524 | 0.503 | 0.516 | 0.505 | 0.513 |
| ETTh2 | 96 | 0.296 | 0.362 | 0.362 | 0.417 | 0.300 | 0.385 | 0.340 | 0.374 | 0.356 | 0.413 | 0.383 | 0.420 | 0.894 | 0.671 | 1.101 | 0.802 | 0.653 | 0.596 |
| | 192 | 0.406 | 0.427 | 0.405 | 0.432 | 0.453 | 0.496 | 0.402 | 0.414 | 0.409 | 0.439 | 0.409 | 0.436 | 0.886 | 0.666 | 0.780 | 0.647 | 0.834 | 0.684 |
| | 336 | 0.513 | 0.498 | 0.750 | 0.625 | 0.705 | 0.610 | 0.452 | 0.452 | 0.413 | 0.441 | 0.389 | 0.435 | 1.115 | 0.764 | 0.963 | 0.762 | 0.932 | 0.711 |
| | 720 | 0.875 | 0.680 | 1.107 | 0.775 | 1.223 | 0.825 | 0.462 | 0.468 | 0.485 | 0.486 | 0.460 | 0.476 | 1.197 | 0.815 | 1.468 | 0.950 | 1.716 | 1.022 |
| ETTm1 | 96 | 0.316 | 0.368 | 0.305 | 0.354 | 0.307 | 0.352 | 0.338 | 0.375 | 0.335 | 0.375 | 0.340 | 0.375 | 0.320 | 0.373 | 0.332 | 0.380 | 0.335 | 0.386 |
| | 192 | 0.374 | 0.404 | 0.353 | 0.390 | 0.354 | 0.386 | 0.374 | 0.387 | 0.448 | 0.429 | 0.436 | 0.425 | 0.427 | 0.451 | 0.375 | 0.422 | 0.383 | 0.429 |
| | 336 | 0.384 | 0.412 | 0.382 | 0.405 | 0.398 | 0.422 | 0.410 | 0.411 | 0.430 | 0.435 | 0.415 | 0.432 | 0.489 | 0.480 | 0.438 | 0.450 | 0.424 | 0.442 |
| | 720 | 0.467 | 0.467 | 0.445 | 0.442 | 0.446 | 0.447 | 0.478 | 0.450 | 0.464 | 0.456 | 0.462 | 0.461 | 0.589 | 0.545 | 0.595 | 0.557 | 0.598 | 0.545 |
| ETTm2 | 96 | 0.179 | 0.275 | 0.177 | 0.278 | 0.193 | 0.283 | 0.187 | 0.267 | 0.184 | 0.272 | 0.191 | 0.277 | 0.353 | 0.424 | 0.459 | 0.463 | 0.361 | 0.413 |
| | 192 | 0.307 | 0.376 | 0.273 | 0.355 | 0.248 | 0.321 | 0.249 | 0.309 | 0.240 | 0.309 | 0.252 | 0.322 | 0.531 | 0.513 | 0.894 | 0.676 | 0.839 | 0.647 |
| | 336 | 0.325 | 0.388 | 0.356 | 0.383 | 0.295 | 0.353 | 0.321 | 0.351 | 0.305 | 0.349 | 0.318 | 0.363 | 0.868 | 0.722 | 0.811 | 0.694 | 1.175 | 0.731 |
| | 720 | 0.502 | 0.490 | 0.433 | 0.445 | 0.427 | 0.447 | 0.408 | 0.403 | 0.400 | 0.407 | 0.390 | 0.409 | 1.880 | 1.010 | 2.052 | 1.098 | 1.319 | 0.907 |

## C  SUPPLEMENT OF RESULTS

### C.1  RANDOMNESS TEST

The parameters of deep neural networks are typically generated through random initialization, which means that different random seeds may lead to different convergence results for the same training set. In this section, we study the impact of different random seeds on the performance of Swin4TS. As a convention, the results of the main text select the current year 2023 as the random seed. In

addition to this, we add four random seeds: 2022, 2021, 2020, and 2019, and evaluate them on 32 tasks across all eight datasets.

In Fig. 7, corresponding to the results of Swin4TS/CI , the influence of random seeds on prediction accuracy is subtle in almost all datasets. Particularly for the Traffic, Electricity, ETTh1, and ETTh2 datasets, the prediction accuracy hardly varies with random seeds. Moreover, Swin4TS/CD is evaluated across 4 ETT datasets (the other 4 datasets are ignored due to the limited time and computer resources) and the results are shown in Fig. 8. Compared to Swin4TS/CI, the random seed seems to have a slightly larger impact on the prediction accuracy of the 4 ETT datasets. However, these effects are still very small in an overall view. All these experiments indicate that the proposed Swin4TS demonstrates strong robustness to different random seeds or model initialization.

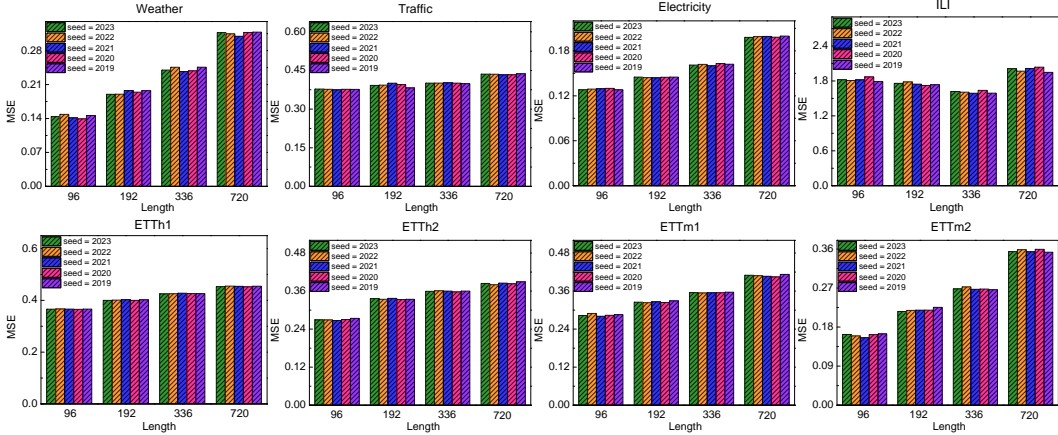

Figure 7: Impact of different random seeds on the performance of Swin4TS/CI

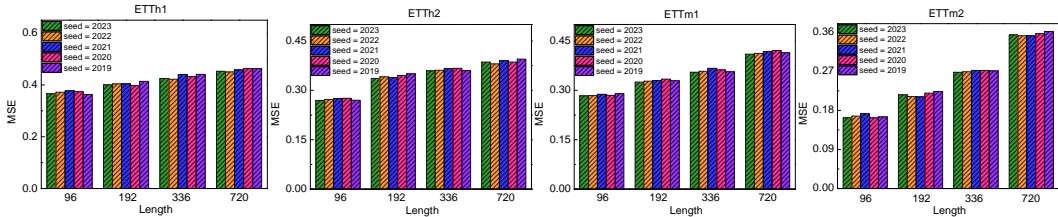

Figure 8: Impact of different random seeds on the performance of Swin4TS/CD

## C.2 ABLATION OF SWIN4TS/CI

As a supplement to the ablation of Swin4TS/CD in the main text, we also conduct experiments on Swin4TS/CI using the ETTm1 and ETTm2 datasets. As shown in Table 11, whether the model removes the shift window attention operation or the hierarchical operation (referred as "scale" here), or removes both, it will result in a decrease in prediction accuracy. Particularly, removing the hierarchical operation leads to a decrease of 4.1% and 2.8% in average prediction accuracy for the ETTm1 dataset (0.341→0.355) and ETTm2 dataset (0.250→0.257), respectively. This indicates that these two key designs in Swin4TS play important roles in ensuring prediction accuracy.

## C.3 EFFECT OF CHANNEL ORDER

For vision models, patches in the window represent adjacent areas in the image. However, when using the CD (channel dependent) strategy on multiple time series data, the channel order cannot be arbitrarily fixed as in images. To learn attention that is independent of channel order, we randomly shuffle the sequence of time series data in the channel dimension at each training batch and shuffle the corresponding labels accordingly in our code implementation.

Table 11: Ablation of Swin4TS/CI on ETTm1 and ETTm2 datasets. Bold/underline indicates the best/second. Prediction length $T \in \{96, 192, 336, 720\}$.

| Models | | Swin4TS/CI | | w/o shift | | w/o scale | | w/o shift&scale | |
|---|---|---|---|---|---|---|---|---|---|
| Metrics | | MSE | MAE | MSE | MAE | MSE | MAE | MSE | MAE |
| ETTm1 | 96 | **0.283** | 0.341 | 0.285 | **0.340** | 0.308 | 0.357 | 0.286 | 0.342 |
| | 192 | **0.325** | **0.366** | 0.326 | 0.367 | 0.335 | 0.372 | 0.330 | 0.368 |
| | 336 | **0.355** | **0.383** | 0.359 | 0.387 | 0.356 | 0.387 | 0.361 | 0.386 |
| | 720 | **0.401** | **0.413** | 0.412 | 0.421 | 0.420 | 0.418 | 0.414 | 0.417 |
| | Avg. | **0.341** | **0.376** | 0.345 | 0.379 | 0.355 | 0.383 | 0.348 | 0.378 |
| ETTm2 | 96 | **0.163** | **0.251** | 0.164 | 0.253 | 0.168 | 0.256 | 0.174 | 0.263 |
| | 192 | **0.216** | **0.292** | 0.225 | 0.298 | 0.221 | 0.293 | 0.224 | 0.296 |
| | 336 | **0.268** | **0.323** | 0.269 | 0.325 | 0.278 | 0.331 | 0.276 | 0.330 |
| | 720 | **0.354** | **0.378** | 0.367 | 0.386 | 0.363 | 0.383 | 0.364 | 0.385 |
| | Avg. | **0.250** | **0.311** | 0.256 | 0.315 | 0.257 | 0.316 | 0.260 | 0.318 |

Table 12 shows the experimental results of shuffling channels and fixing channels, and the results show that this technique achieved better results than fixing the initial order. This indicates that under the CD strategy, the model can learn interrelationship among multiple series that is independent of channel order. In other words, the order of time series in the channel dimension is insignificant in ideal situations.

Table 12: Effect of channel order under the CD strategy .

| Datasets | | ETTh1 | | | | ETTh2 | | | | ETTm1 | | | | ETTm2 | | | |
|---|---|---|---|---|---|---|---|---|---|---|---|---|---|---|---|---|---|
| prediction | | 96 | 192 | 336 | 720 | 96 | 192 | 336 | 720 | 96 | 192 | 336 | 720 | 96 | 192 | 336 | 720 |
| Channel-fixed | MSE | 0.382 | 0.414 | 0.435 | 0.49 | 0.286 | 0.35 | 0.375 | 0.417 | **0.288** | 0.34 | 0.375 | 0.441 | 0.165 | 0.229 | 0.279 | 0.365 |
| | MAE | 0.407 | 0.431 | 0.451 | 0.492 | 0.351 | 0.400 | 0.422 | 0.448 | **0.341** | 0.37 | 0.39 | 0.427 | 0.252 | 0.298 | 0.329 | 0.385 |
| Channel-shuffled | MSE | **0.365** | **0.400** | **0.425** | **0.432** | **0.264** | **0.331** | **0.358** | **0.386** | 0.292 | **0.334** | **0.364** | **0.402** | **0.160** | **0.219** | **0.268** | **0.344** |
| | MAE | **0.392** | **0.414** | **0.44** | **0.456** | **0.33** | **0.375** | **0.401** | **0.427** | 0.346 | **0.369** | **0.387** | **0.417** | **0.249** | **0.29** | **0.322** | **0.371** |

## C.4 EFFECT OF HISTORICAL SERIES LENGTH

The full results considering the effect of $L$ of Swin4TS are shown in Table 13. Overall, the best result for each prediction task occur in a longer $L$, which is $L = 512$ for most cases. Even in many tasks for ETT datasets, although $L = 640$ can provide the best results, the difference from the second-best results obtained with $L = 512$ is negligible. Additionally, due to the linear computational complexity of Swin4TS with respect to $L$, it is inherently more suitable for processing longer $L$ compared to other baselines. It is worth noting that even with $L = 96$, the prediction results of Swin4TS are better than those of most baselines.

## C.5 EFFECT OF HIERARCHICAL DESIGN

The full results considering the effect of hierarchical design of Swin4TS are shown in Table 14. As mentioned in the main text, different hierarchical designs capture different scale information hidden in the time series. For Weather and 4 ETT datasets, a 2-stage design yields the best performance. On the other hand, the Traffic and Electricity datasets require a 4-stage design to achieve optimal performance. Additionally, the impact of different stage designs on the same dataset varies much. For example, for Weather dataset, the four hierarchical designs show minor differences in prediction results. However, for ETTm2 dataset, switching from a 2-stage to a 4-stage design leads to a 22.5% decrease in average prediction performance (0.253→0.309).

## C.6 EFFECT OF DYNAMIC COVARIATES

In many time series prediction scenarios, the design of the model often considers known auxiliary covariates, such as timestamps, or specific time series. As a test, the dynamic covariates are incorporated into the Swin4TS/CI model. Specifically, given a look-back window $L : \mathbf{X} \in \mathbb{R}^{1 \times L}$ of time series, we obtain the same length of timestamp $\mathbf{X}_c \in \mathbb{R}^{1 \times L}$, map the time covariates to a low-dimensional space $\mathbf{X}_c \in \mathbb{R}^{d \times L}$, where $d$ represents the dimensional space of dynamic time covariates. We then concatenate the mapped time covariates $\mathbf{X}_c$ with the time variable $\mathbf{X}$ and input them together into the Swin4TS/CI model.

Table 13: Full results for the study of historical sequence length $L$. The best results are bolded. Prediction length $T \in \{24, 36, 48, 60\}$ for ILI dataset and $T \in \{96, 192, 336, 720\}$ for the others. And $L \in \{36, 54, 81, 108, 135\}$ for ILI dataset and $L \in \{96, 128, 256, 512, 720\}$ for the others.

| Models | | 96(36) | | 128(54) | | 256(81) | | 512(108) | | 640(135) | |
|---|---|---|---|---|---|---|---|---|---|---|---|
| Metrics | | MSE | MAE | MSE | MAE | MSE | MAE | MSE | MAE | MSE | MAE |
| Weather | 96 | 0.177 | 0.218 | 0.162 | 0.205 | 0.152 | 0.196 | **0.143** | **0.191** | 0.146 | 0.196 |
| | 192 | 0.218 | 0.254 | 0.207 | 0.246 | 0.196 | 0.239 | 0.189 | 0.235 | **0.188** | **0.236** |
| | 336 | 0.274 | 0.294 | 0.263 | 0.287 | 0.248 | 0.280 | **0.238** | **0.278** | 0.241 | 0.279 |
| | 720 | 0.353 | 0.345 | 0.342 | 0.340 | 0.328 | 0.335 | **0.312** | **0.328** | 0.325 | 0.750 |
| Traffic | 96 | 0.485 | 0.308 | 0.429 | 0.283 | 0.401 | 0.270 | **0.362** | **0.249** | 0.376 | 0.265 |
| | 192 | 0.482 | 0.304 | 0.448 | 0.290 | 0.413 | 0.273 | **0.378** | **0.255** | 0.389 | 0.270 |
| | 336 | 0.500 | 0.311 | 0.458 | 0.294 | 0.425 | 0.279 | **0.389** | **0.261** | 0.402 | 0.277 |
| | 720 | 0.531 | 0.328 | 0.490 | 0.312 | 0.453 | 0.297 | **0.426** | **0.282** | 0.437 | 0.293 |
| Electricity | 96 | 0.178 | 0.262 | 0.155 | 0.246 | 0.138 | 0.232 | **0.127** | **0.221** | 0.131 | 0.227 |
| | 192 | 0.184 | 0.268 | 0.170 | 0.258 | 0.153 | 0.245 | **0.144** | **0.237** | 0.147 | 0.242 |
| | 336 | 0.200 | 0.285 | 0.185 | 0.274 | 0.169 | 0.262 | **0.16** | **0.255** | 0.164 | 0.259 |
| | 720 | 0.241 | 0.319 | 0.225 | 0.308 | 0.210 | 0.298 | **0.197** | **0.288** | 0.199 | 0.292 |
| ILI | 24 | 2.525 | 0.944 | **1.716** | **0.813** | 1.912 | 0.855 | 1.740 | 0.854 | 1.854 | 0.892 |
| | 36 | 2.953 | 0.971 | 1.901 | **0.855** | 1.823 | 0.864 | **1.713** | 0.859 | 1.971 | 0.935 |
| | 48 | 2.431 | 0.929 | 1.863 | **0.850** | 2.102 | 0.948 | **1.718** | 0.877 | 2.048 | 0.956 |
| | 60 | 1.951 | **0.858** | 2.049 | 0.954 | 2.115 | 0.971 | **1.790** | 0.908 | 2.117 | 0.975 |
| ETTh1 | 96 | 0.384 | 0.395 | 0.383 | 0.395 | 0.376 | 0.394 | **0.366** | **0.394** | **0.366** | 0.395 |
| | 192 | 0.435 | 0.424 | 0.433 | 0.422 | 0.414 | **0.416** | **0.403** | **0.420** | 0.401 | 0.417 |
| | 336 | 0.478 | 0.445 | 0.469 | 0.440 | 0.440 | **0.431** | **0.425** | 0.433 | 0.430 | 0.437 |
| | 720 | 0.481 | 0.468 | 0.474 | 0.463 | 0.454 | **0.461** | **0.448** | 0.463 | 0.462 | 0.477 |
| ETTh2 | 96 | 0.288 | 0.338 | 0.288 | 0.339 | 0.276 | 0.336 | 0.272 | 0.334 | **0.268** | **0.333** |
| | 192 | 0.372 | 0.389 | 0.339 | 0.388 | 0.345 | 0.381 | 0.336 | 0.377 | **0.334** | **0.376** |
| | 336 | 0.413 | 0.425 | 0.404 | 0.422 | 0.371 | 0.406 | 0.362 | 0.404 | **0.358** | **0.396** |
| | 720 | 0.420 | 0.440 | 0.415 | **0.415** | 0.396 | 0.429 | **0.384** | 0.427 | 0.389 | 0.431 |
| ETTm1 | 96 | 0.326 | 0.359 | 0.304 | 0.350 | 0.290 | 0.345 | 0.283 | 0.341 | **0.282** | **0.339** |
| | 192 | 0.362 | 0.380 | 0.345 | 0.375 | 0.325 | 0.369 | 0.325 | **0.366** | **0.323** | 0.367 |
| | 336 | 0.389 | 0.399 | 0.376 | 0.396 | 0.357 | 0.388 | 0.355 | **0.383** | **0.353** | 0.388 |
| | 720 | 0.452 | 0.435 | 0.437 | 0.432 | 0.418 | 0.426 | 0.401 | **0.413** | **0.399** | 0.417 |
| ETTm2 | 96 | 0.175 | 0.260 | 0.174 | 0.260 | 0.164 | 0.252 | 0.163 | **0.251** | **0.161** | **0.251** |
| | 192 | 0.241 | 0.302 | 0.234 | 0.299 | 0.221 | 0.293 | **0.216** | 0.292 | **0.216** | **0.290** |
| | 336 | 0.301 | 0.340 | 0.293 | 0.339 | 0.277 | 0.329 | **0.268** | **0.323** | 0.269 | 0.325 |
| | 720 | 0.399 | 0.396 | 0.389 | 0.392 | 0.369 | 0.387 | 0.354 | **0.378** | **0.352** | **0.378** |

The experimental results are shown in Table 15. As can be seen, the performances of most datasets become worse after adding this covariate, with the exception of the Traffic dataset which performed much better, possibly due to the strong periodicity of traffic flow with time.

## C.7 TRANSFERABILITY OF SWIN4TS/CI

In Fig. 9, we further test the effect when one of these 7 channels is fixed (instead of randomly chosen) in the training stage of Swin4TS/CI. For ETTh1 and ETTm1, fixing one of the first 4 channels for training, the prediction accuracy does not change much, suggesting that these 4 channels can globally capture the underlying characteristics of the whole dataset. While fixing one of the left 3 channels, the prediction accuracy increases by a large margin. For ETTh2 and ETTm2, fixing any one of these 7 channels can impact the prediction accuracy, but the fluctuations of impacts are not substantial, especially for ETTm2. In fact, this experiment can be considered as a test of the transferability for Swin4TS. That is, the model trained on data from one channel can still perform well in predicting unseen data from other channels.

## D A U-NET DESIGN FOR SWIN4TS/CD

In Swin4TS/CD, the final layer is mapped to $MT$ (where $M$ represents the number of series and $T$ denotes the prediction length) by a linear layer which keeps the consistency with Swin4TS/CI. However, this may not be a good choice especially for large datasets where the size of last layer would become very large for long-term prediction. To address this issue, we have additionally designed a U-net architecture to consider the problem in the last layer. The overall structure of the network is shown in Fig. 10, which consists of a symmetrical Encoder-Decoder structure. The

Table 14: Full results for the study of hierarchical design. Four cases, representing 1-stage, 2-stage, 3-stage and 4-stage, are studied across 8 datasets. Prediction length $T \in \{24, 36, 48, 60\}$ for ILI dataset and $T \in \{96, 192, 336, 720\}$ for the others.

| Models | | 1-stage | | 2-stage | | 3-stage | | 4-stage | |
|---|---|---|---|---|---|---|---|---|---|
| Metrics | | MSE | MAE | MSE | MAE | MSE | MAE | MSE | MAE |
| Weather | 96 | 0.149 | 0.199 | **0.143** | **0.191** | 0.146 | 0.196 | 0.145 | 0.195 |
| | 192 | 0.192 | 0.240 | **0.189** | **0.235** | 0.191 | 0.238 | 0.191 | 0.239 |
| | 336 | 0.244 | 0.281 | **0.238** | 0.278 | 0.244 | 0.279 | 0.241 | **0.277** |
| | 720 | 0.316 | 0.331 | **0.312** | **0.328** | 0.318 | 0.332 | 0.317 | 0.333 |
| Traffic | 96 | 0.381 | 0.261 | 0.378 | 0.263 | 0.368 | 0.254 | **0.362** | **0.249** |
| | 192 | 0.394 | 0.266 | 0.392 | 0.268 | 0.384 | 0.260 | **0.378** | **0.255** |
| | 336 | 0.404 | 0.271 | 0.401 | 0.272 | 0.394 | 0.265 | **0.389** | **0.261** |
| | 720 | 0.439 | 0.290 | 0.436 | 0.291 | 0.431 | 0.286 | **0.426** | **0.282** |
| Electricity | 96 | 0.132 | 0.227 | 0.131 | 0.226 | 0.128 | 0.223 | **0.127** | **0.221** |
| | 192 | 0.148 | 0.240 | 0.147 | 0.241 | 0.145 | 0.238 | **0.144** | **0.237** |
| | 336 | 0.164 | 0.257 | 0.163 | 0.258 | 0.161 | 0.256 | **0.16** | **0.255** |
| | 720 | 0.202 | 0.290 | 0.201 | 0.292 | 0.198 | 0.289 | **0.197** | **0.288** |
| ILI | 24 | **1.82** | **0.873** | 1.874 | 0.883 | 1.901 | 0.848 | - | - |
| | 36 | 1.758 | 0.873 | **1.703** | **0.849** | 2.382 | 1.045 | - | - |
| | 48 | **1.619** | **0.851** | 2.205 | 1.013 | 2.133 | 0.973 | - | - |
| | 60 | **2.011** | **0.938** | 2.088 | 0.960 | 2.137 | 2.137 | - | - |
| ETTh1 | 96 | 0.369 | 0.395 | **0.366** | 0.394 | 0.367 | **0.393** | 0.376 | 0.404 |
| | 192 | 0.401 | 0.413 | 0.403 | **0.42** | **0.401** | 0.421 | 0.42 | 0.435 |
| | 336 | **0.425** | 0.444 | **0.425** | **0.433** | 0.427 | 0.438 | 0.45 | 0.458 |
| | 720 | **0.445** | 0.468 | 0.448 | **0.463** | 0.475 | 0.485 | 0.474 | 0.478 |
| ETTh2 | 96 | 0.273 | 0.336 | **0.272** | **0.334** | 0.285 | 0.345 | 0.318 | 0.373 |
| | 192 | 0.337 | 0.380 | **0.336** | **0.377** | 0.346 | 0.383 | 0.369 | 0.407 |
| | 336 | 0.361 | **0.402** | 0.362 | 0.404 | 0.368 | 0.404 | 0.382 | 0.422 |
| | 720 | 0.390 | 0.432 | **0.384** | **0.427** | 0.392 | 0.430 | 0.414 | 0.449 |
| ETTm1 | 96 | 0.285 | 0.342 | 0.283 | 0.341 | **0.282** | **0.340** | 0.289 | 0.346 |
| | 192 | 0.331 | 0.371 | 0.325 | **0.366** | **0.323** | 0.367 | 0.329 | 0.371 |
| | 336 | 0.371 | 0.395 | **0.355** | **0.383** | **0.355** | 0.387 | 0.369 | 0.39 |
| | 720 | 0.407 | 0.418 | **0.401** | **0.413** | 0.404 | 0.419 | 0.414 | 0.424 |
| ETTm2 | 96 | 0.165 | 0.254 | **0.163** | **0.251** | 0.179 | 0.267 | 0.232 | 0.317 |
| | 192 | 0.220 | **0.292** | **0.216** | 0.292 | 0.240 | 0.309 | 0.271 | 0.338 |
| | 336 | 0.270 | 0.326 | **0.268** | **0.323** | 0.291 | 0.342 | 0.327 | 0.373 |
| | 720 | 0.358 | 0.381 | **0.354** | **0.378** | 0.384 | 0.402 | 0.405 | 0.419 |

Table 15: Effect of dynamic covariates under the CI strategy.

| Datasets | | ETTh1 | | | | ETTh2 | | | | ETTm1 | | | | ETTm2 | | | |
|---|---|---|---|---|---|---|---|---|---|---|---|---|---|---|---|---|---|
| prediction | | 96 | 192 | 336 | 720 | 96 | 192 | 336 | 720 | 96 | 192 | 336 | 720 | 96 | 192 | 336 | 720 |
| w/timestamp | MSE | 0.371 | 0.408 | 0.436 | 0.5 | 0.281 | 0.35 | 0.372 | 0.401 | 0.291 | 0.342 | 0.409 | 0.46 | 0.171 | 0.229 | 0.297 | 0.418 |
| | MAE | 0.400 | 0.423 | 0.446 | 0.498 | 0.345 | 0.389 | 0.408 | 0.442 | 0.348 | 0.38 | 0.424 | 0.446 | 0.26 | 0.301 | 0.346 | 0.425 |
| w.o/timestamp | MSE | **0.366** | **0.400** | **0.425** | **0.448** | **0.269** | **0.336** | **0.359** | **0.384** | **0.283** | **0.325** | **0.355** | **0.401** | **0.163** | **0.216** | **0.268** | **0.354** |
| | MAE | **0.392** | **0.416** | **0.433** | **0.463** | **0.334** | **0.377** | **0.400** | **0.427** | **0.341** | **0.366** | **0.385** | **0.415** | **0.251** | **0.292** | **0.323** | **0.378** |

| Datasets | | Weather | | | | Traffic | | | | Electricity | | | | ILI | | | |
|---|---|---|---|---|---|---|---|---|---|---|---|---|---|---|---|---|---|
| prediction | | 96 | 192 | 336 | 720 | 96 | 192 | 336 | 720 | 96 | 192 | 336 | 720 | 96 | 192 | 336 | 720 |
| w/timestamp | MSE | 0.149 | 0.195 | 0.246 | 0.318 | **0.329** | **0.351** | **0.367** | **0.377** | 0.131 | 0.147 | 0.172 | 0.212 | 1.818 | 1.948 | 1.884 | 1.904 |
| | MAE | 0.200 | 0.242 | 0.281 | 0.333 | **0.241** | **0.252** | **0.259** | **0.266** | 0.231 | 0.246 | 0.273 | 0.303 | 0.867 | 0.924 | 0.925 | 0.945 |
| w.o/timestamp | MSE | **0.144** | **0.190** | **0.240** | **0.312** | 0.362 | 0.378 | 0.389 | 0.426 | **0.127** | **0.144** | **0.160** | **0.197** | **1.74** | **1.713** | **1.718** | **1.79** |
| | MAE | **0.192** | **0.236** | **0.278** | **0.329** | 0.249 | 0.255 | 0.261 | 0.282 | **0.221** | **0.237** | **0.255** | **0.288** | **0.854** | **0.859** | **0.877** | **0.908** |

Figure 9: Swin4TS/CI with one fixed channel as training data. Four ETT datasets are considered. Prediction length $T \in \{96, 192, 336, 720\}$.

left half of the network is exactly the same as Swin4TS/CD, while the right half shares the same framework only replacing the downsampling layer with upsampling layer. Such a U-net architecture is quite similar to the one in CV for image generation tasks, and the decoder gradually recovers the target multi-series. Additionally, we included residual connections to the attention models to better utilize the information from the encoder network.

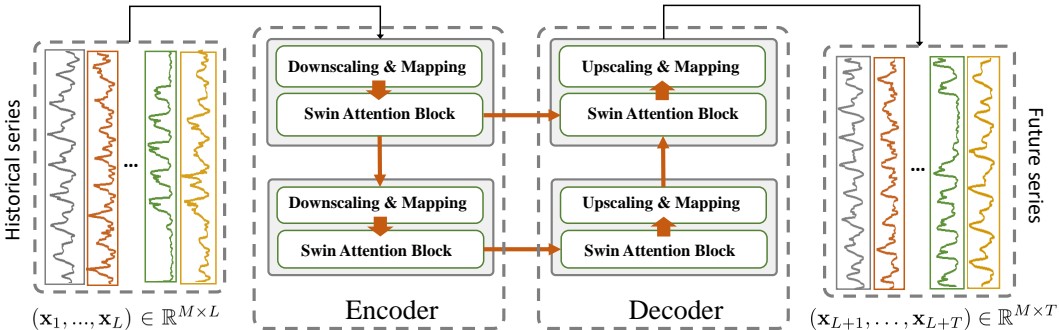

Figure 10: The overall structure of the U-net design for Swin4TS/CD.

We have performed experiments on four datasets: Traffic, Electricity, ETTh1 and ETTh2, and the results are shown in Table 16. As can be seen, the U-net design outperforms the Linear design in handling datasets with a large number of channels, such as Traffic and Electricity, indicating the efficiency in addressing challenges in modeling the multi-variables correlation for long-term prediction. In consideration of the simplicity and consistency with the CI strategy, the last layer of Swin4TS/CD in the main text still adopts the simple linear design.

Table 16: Comparison of the different designs of last layer for Swin4TS/CD. Best results are bold.

| Datasets | | Traffic | | | | Electricity | | | | ETTh1 | | | | ETTh2 | | | |
|---|---|---|---|---|---|---|---|---|---|---|---|---|---|---|---|---|---|
| prediction | | 96 | 192 | 336 | 720 | 96 | 192 | 336 | 720 | 96 | 192 | 336 | 720 | 96 | 192 | 336 | 720 |
| Linear | MSE | 0.512 | 0.522 | 0.529 | 0.541 | 0.159 | 0.170 | 0.175 | 0.201 | **0.365** | 0.400 | **0.425** | **0.432** | **0.264** | **0.331** | **0.358** | **0.386** |
| | MAE | 0.325 | 0.333 | 0.341 | 0.343 | 0.271 | 0.268 | 0.282 | 0.299 | **0.392** | 0.414 | **0.440** | 0.456 | **0.330** | **0.375** | **0.401** | **0.427** |
| U-net | MSE | **0.478** | **0.482** | **0.522** | **0.518** | **0.142** | **0.158** | **0.166** | **0.187** | 0.371 | **0.398** | 0.432 | 0.433 | 0.277 | 0.352 | 0.366 | 0.389 |
| | MAE | **0.298** | **0.302** | **0.314** | **0.320** | **0.243** | **0.255** | **0.265** | **0.283** | 0.401 | **0.412** | 0.444 | **0.454** | 0.345 | 0.393 | 0.416 | 0.441 |

# E  TEST OF THE TNT4TS MODEL

In the main text, we addressed the consistency in data structure between the time series and image modalities, thus enabling the processing of these two modalities in a unified framework. To further support this view, we designed the TNT4TS model in addition. TNT (Transformer in Transformer) is an early ViT model that examines the internal structure of images, using two Transformer models to process local and global information, respectively. TNT4TS is an attempt on TNT to model time series. As shown in Fig. 11, time series are first divided into multiple patches, and then each patch is further divided into multiple sub-patches. TNT4TS constructs an Outer Transformer to process the patch series, and then uses an Inner Transformer to locally process the sub-patch series of each patch. The information from the Inner Transformer is fused into the Outer Transformer, and finally the output of the latter is mapped to the predicted sequence via a Linear layer. The main difference between TNT4TS and TNT is just that the former processes one-dimensional information while the latter processes two-dimensional information. For detailed framework design, please refer to the original paper of TNT.

Using the CI strategy, we evaluate the performance of TNT4TS on 4 ETT datasets and compared it with Swin4TS/CI and PatchTST. Results are presented in Table 17. As can be seen, overall, TNT4TS shows comparable performance with Swin4TS/CI, but outperforms PatchTST on all datasets. This further demonstrates the feasibility of using the ViT framework for time series modeling. We look forward to seeing more advanced ViT models being used for time series modeling in the future.

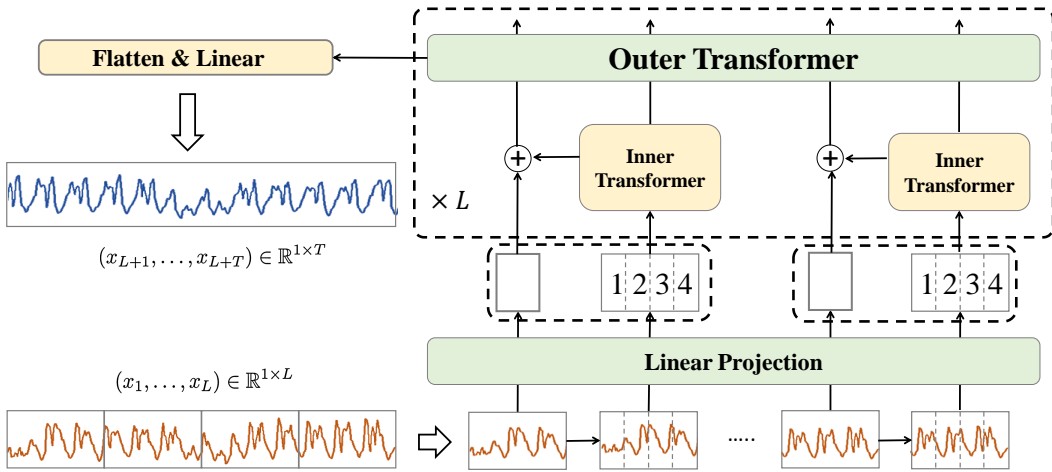

Figure 11: The overall structure of the TNT4TS model.

Table 17: The evaluation of TNT4TS on 4 ETT datasets. Best results are bold.

| Datasets | | ETTh1 | | | | ETTh2 | | | | ETTm1 | | | | ETTm2 | | | |
|---|---|---|---|---|---|---|---|---|---|---|---|---|---|---|---|---|---|
| prediction | | 96 | 192 | 336 | 720 | 96 | 192 | 336 | 720 | 96 | 192 | 336 | 720 | 96 | 192 | 336 | 720 |
| Swin4TS/CI | MSE | 0.366 | **0.403** | 0.425 | 0.448 | 0.272 | 0.336 | 0.362 | 0.384 | **0.283** | **0.325** | **0.355** | **0.401** | **0.163** | **0.216** | **0.268** | 0.354 |
| | MAE | 0.394 | **0.42** | 0.433 | 0.463 | 0.334 | **0.377** | 0.404 | 0.427 | **0.341** | **0.366** | **0.383** | **0.413** | **0.251** | 0.292 | **0.323** | 0.378 |
| TNT4TS | MSE | **0.363** | **0.403** | **0.416** | **0.445** | **0.269** | **0.335** | **0.360** | **0.378** | 0.288 | 0.335 | 0.362 | 0.409 | 0.164 | 0.218 | 0.268 | 0.355 |
| | MAE | **0.390** | 0.422 | **0.427** | **0.461** | **0.328** | 0.378 | **0.401** | **0.423** | 0.348 | 0.381 | 0.388 | 0.422 | 0.254 | **0.292** | 0.327 | **0.377** |
| PatchTST | MSE | 0.377 | 0.411 | 0.432 | 0.456 | 0.275 | 0.339 | 0.365 | 0.39 | 0.292 | 0.331 | 0.367 | 0.421 | 0.166 | 0.221 | 0.271 | 0.361 |
| | MAE | 0.405 | 0.428 | 0.445 | 0.473 | 0.339 | 0.38 | 0.404 | 0.43 | 0.346 | 0.37 | 0.391 | 0.42 | 0.256 | 0.294 | 0.327 | 0.384 |

## F  PREDICT EXAMPLES

Here we present a comparison of the actual prediction results of Swin4TS/CI with several representative baselines. As shown in Fig. 12, the four columns represent Swin4TS/CI, PatchTST, MICN, and TimesNet. The four rows correspond to the ETTh1, ETTm1, ETTm2, and Weather datasets, predicting 192, 192, 336, and 336 future values, respectively.

For the first prediction scenario, ETTh1 dataset, Swin4TS/CI and TimesNet provide reasonable predictions, but PatchTST exhibits excessive fluctuations, and MICN shows significant deviations in the prediction trends. In the second scenario, ETTm1 dataset, all four algorithms seem to have less accurate predictions in the first half, but Swin4TS/CI and TimesNet give accurate forecasts in the second half. In the third scenario, ETTm2 dataset, where the ground truth of predicted part demonstrates a downward trend, Swin4TS/CI and PatchTST accurately predict this trend, although PatchTST has a larger deviation in predicting the second peak. Meanwhile, MICN and TimesNet fail to forecast this downward trend. In the fourth scenario, Weather, Swin4TS/CI and PatchTST can provide fairly accurate trends, while MICN significantly overestimates the predicted values, and TimesNet underestimates the predicted values (especially the second peak). Overall, PatchTST, MICN, and TimesNet perform well only in certain scenarios, while Swin4TS/CI can provide relatively accurate forecasts across all scenarios.

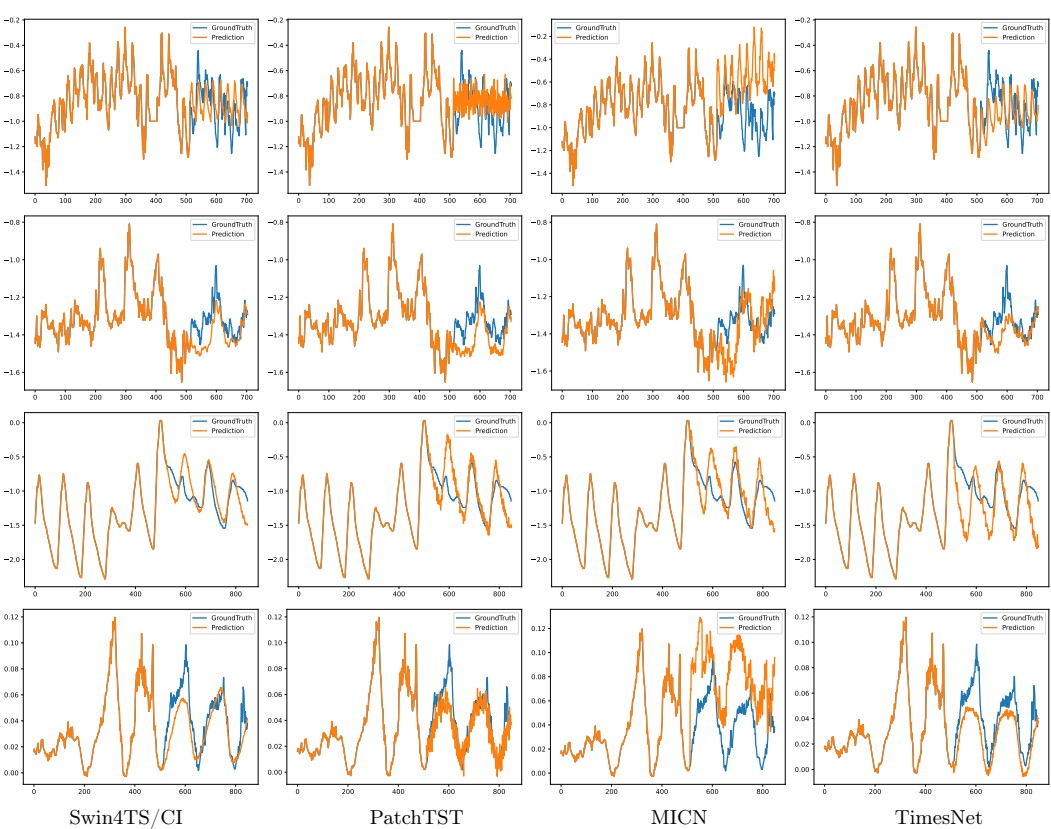

Figure 12: Predicted examples of ETTh1, ETTm1, ETTm2 and Weather datasets, corresponds to the results of four rows, respectively.

