# OpenReview forum: "Long-term Time Series Forecasting with Vision Transformer"
_ICLR.cc/2024/Conference — Submitted to ICLR 2024_

### Official Review · Reviewer_SUuV · 2023-10-20

**Soundness:** 2 fair
**Presentation:** 2 fair
**Contribution:** 2 fair
**Rating:** 5
**Confidence:** 4

**Summary:**

The authors propose a model called Swin4TS for long-term time series forecasting. Swin4TS adopts SwinTransformer with native window-based attention and hierarchical representations adapted for the time series modality, which has shown powerful capability in CV. The paper provides two variants, Swin4TS/CD and Swin4TS/CI, which can adapt to channel-dependent and channel-independent strategies, respectively. The experiments show that the model outperforms existing baselines on eight benchmark datasets.

**Strengths:**

1. Swin4TS achieves improved performance on multiple benchmark datasets.

**Weaknesses:**

1. The paper is somewhat not well-organized. The figures (1, 2, and 4) are not rough to convey enough insights and are unportable. The tables (2, 3, and 4) for many experiments are incomplete on the dataset. Some notations can be not so rigorous (\hat_x may be better to denote the model predictions). Thus, this paper essentially needs further polishing.
2. The contributions of the paper can be not enough, which is the most fatal point. Swin-transformer modules are adapted for the time series. It's like lowering the dimension of attention directly on 1D time series representations. Is this more motivating than design models considering time series properties? How does inductive bias like shift invariance help time series forecasting? Without considering the gap between the two modalities, I think it has not contributed much to the research field.
3. The performance incremental is marginal. The forecasting result of Swin4TS over PatchTST can be small. And I think the ablations should be conducted on more extensive datasets. Also, I suggest that experiments such as hyperparameter effects are more suitable to be placed in the appendix, and more experiments exploring the helpful inductive bias should be considered.
4. Unsolved contribution: "We successfully apply techniques from ViT to LTSF", which does not "indicate the potential of connecting time series with other domains within the Transformer architecture". What does the 'connection' mean? The author provides no experiments to validate the results of cross-domain/modality transfer.

**Questions:**

1. SwinTransformer in computer vision are commonly leveraged for high-level feature extraction, which is widely applied on tasks such as image classification. The time series forecasting task, on the contrary, belongs to a typical generative task and needs to learn low-level feature representation for time point reconstruction. How does the proposed model cope with the gap between tasks and representation granularity?
2. Is it an advantage that being able to combine CI/CD mechanisms? Previous forecasting models are commonly implemented by CD, but they can also be easily incorporated with CI, no matter Transformers or non-Transformers.

---

> ### Author Response · Authors · 2023-11-18
> **Response to reviewer SUuV (Part 1/2)**
>
> &emsp;&emsp;Thank you for your comment. We have carefully and seriously considered the questions and concerns you raised, and provided targeted explanations. We hope our response can change your perception of this work.
>
> ### **For the 1st concern of weaknesses**
> &emsp;&emsp;The reviewer mentioned that Figures 1, 2, and 4 do not clearly convey insights. We do not quite understand. Do you have any clearer suggestions? For Tables 2, 3, and 4. Table 2 shows the results of univariate prediction, for which **we only evaluated on the 4 ETT datasets. This is because in the field of time series prediction, univariate prediction is not the main challenge, and conventionally it is evaluated only on these 4 datasets or less**. Anyway, we evaluated Swin4TS on the other four datasets as shown below (evaluated by MSE), and the results showed that Swin4TS still outperformed other baselines.
> |             | Swin4TS/CI | PatchTST/64 |  MICN  | TimesNet | FEDformer | Autoformer |
> |:-----------:|:----------:|:-----------:|:------:|:--------:|:---------:|------------|
> |   Weather   |    0.002   |    0.002    | 0.013  |  0.002   |   0.014   |   0.058    |
> |   Traffic   |    0.134   |    0.193    | 0.285  |  0.167   |   0.236   |   0.271    |
> | Electricity |    0.277   |    0.709    | 0.364  |  0.422   |   0.373   |   0.421    |
> |     ILI     |    0.756   |    0.749    | 2.658  |  0.818   |   1.079   |   1.079    |
>
> However, this comparison is unfair to other baselines because they have not conducted the same experiments in their original work, and they may require appropriate adjustments in hyperparameters for these additional datasets to achieve optimal performance. Table 3 is the ablation analysis for ETTm1 and ETTm2. We also conducted ablation experiments on Swin4TS/CI on four other datasets, and in most cases, the effectiveness was confirmed. Please understand that due to limited time, we were unable to complete all the ablation experiments.
> |           | Weather | Traffic | Electricity |  ILI  |
> |:---------:|:-------:|:-------:|:-----------:|:-----:|
> | with all  |  0.220  |  0.356  |    0.157    | 1.740 |
> | w/o shift |  0.222  |  0.355  |    0.157    | 1.751 |
> | w/o scale |  0.228  |  0.358  |    0.165    | 1.812 |
> |  w/o both |  0.225  |  0.359  |    0.162    | 1.787 |
>
> **It should be noted that presenting obvious ablation effects for all datasets is demanding. At least for time series prediction, it is customary to present ablation results only on some datasets**. Table 4 is about the experiment on hierarchy design. We do not take the 4-stage experiment for the ILI dataset. This is because the input length of ILI is 108 which is unreasonable to design a 4-stage architecture, as we have mentioned in the main text. **Overall, results of this work seem incomplete, primarily due to some customary practices rather than our work being incomplete. Hope the reviewer can consider again.**
>
> ### **For the 2nd concern**
> &emsp;&emsp; We would like to reiterate the motivation behind our work. Swin Transformer is essentially a Transformer that takes into account information at multiple scales (i.e., hierarchical design) and combines window-based attention to reduce complexity to linear. Although it was originally proposed to solve CV problems, it is essentially designed for handling sequential data. Time series is naturally sequential data, while image is partitioned into patch series. Therefore, from this perspective, Swin Transformer is even more suitable for handling time series.
>
> &emsp;&emsp; **We want to emphasize that the core architecture of Swin Transformer is independent of CV tasks**. It can be applied to a wide range of tasks for modeling sequential data. A good example of this is the Video Swin Transformer (VST, https://arxiv.org/abs/2106.13230), which actually deals with image-based time series. In fact, VST has not made specific designs for video data, but simply extends Swin Transformer from 2D to 3D, and has achieved SOTA performance on different tasks. **If each frame of video data is reduced to a point or vector, then video data becomes time series. This means that VST already implies an attempt to process time series data with Swin Transformer. The main contribution of our work is showing the structural similarity of time series and image, making these two modalities be modeled within a unified framework**.
>
> (To be continued in Part 2/2)

---

> ### Author Response · Authors · 2023-11-18
> **Response to reviewer SUuV (Part 2/2)**
>
> (Following Part 1/2)
>
> &emsp;&emsp; Indeed, time series and images have their own characteristics. Or, as you mentioned, there is a gap between these two modalities. **However, the characteristics of these two modalities can be integrated  to their own underlying patterns. As long as Swin Transformer can learn these underlying patterns, there is no need to design too much specifically for their own features. As you know, ViTs do not have strong inductive bias like CNNs. But they still can get superior performance by training on large datasets**. For example the shift invariance in image you mentioned. Due to that the partition of image in ViT destroys the space structure, ViT uses a positional embedding to learn the space structure. For time series analysis, at least for forecasting task, the shift invariance is not obvious. Thus the positional embedding and the relative bias in Swin Transformer do not facilitate the performance in practice. While for time series classification task, we expect to see the shift invariance may help the model performance.
>
> &emsp;&emsp; **Anyway, thank you for pointing out this. Your concerns made us realize that we did not clearly articulate our motivation and contribution in the previous manuscript. In the Conclusion section of the revised one, we added a short discussion to comprehensively address our views. Please kindly consider again.**
>
> ### **For the 3rd concern**
> &emsp;&emsp; Regarding the performance incremental. **The SOTA for time series prediction has reached a bottleneck**. As far as we know, very recent works nearly all have limited improvements for PatchTST. Regarding the effectiveness of Swin4TS, we have additionally provided more experimental results to validate the feasibility and effectiveness of Swin4TS, such as TNT4TS and hierarchical analysis. We hope that these results will meet your requirements. Regarding the necessity of using backbones from other fields. As stated in our main text, the significance of our work lies in the recognition of the structural similarity between the time series and image modalities, and modeling them in a unified framework. **Therefore, time series modeling can draw on advanced models from the image modality to solve its own problems, instead of designing various sophisticated models as before. This will promote the development of time series analysis**. Finally, the reviewer suggested more helpful inductive bias to prove the rationality of Swin4TS. This is back to the 2nd concern mentioned above, which we have stated.
>
> ###  **For the 4th concern**
> &emsp;&emsp;Our wording may have been misleading. **Our intention was to use a unified model to model time series and image patch series based on their structural similarities, rather than providing contributions for multi-modality task**. In the revised manuscript, we expressed it in a different way.
>
> ###  **For the 1st question**
> &emsp;&emsp;Regarding the gap. When Swin Transformer is applied for image classification, it first obtains a representation of the image, and then uses this representation for classification by a linear layer. Similarly, Swin4TS obtains a representation of the input time series and then directly connects a linear layer to map the future time series. **The main difference is just that the former is 2-dimensional and the latter is 1-dimensional**.
>
> &emsp;&emsp;  Regarding the representation granularity. Since existing time series datasets are far less complete than image datasets, large models may overfit. The size of Swin4TS is smaller than Swin4CV, which is mainly reflected in the smaller numbers of attention heads, layers, and stages and dimension of hidden vectors. However, **this does not mean that the underlying pattern of time series is essentially low-level. The patterns of time series data, such as meteorological data, may be very complex**, especially considering the interrelationships between multiple time series.
>
> ### **For the 2nd question**
> &emsp;&emsp;Previous prediction models, such as Transformer, Autoformer, and Fedformer, can indeed be compatible with CI/CD. However, these models just fuse multiple channels with an embedding layer and do not explicitly consider the correlations between them, and thus the complexity of these models is not relevant to the number of channels. **Swin4TS, on the other hand, can be compatible with both CI/CD strategies and can explicitly consider the correlations between multiple variables under CD strategy**.

---

> ### Comment · Reviewer_SUuV · 2023-11-20
>
> Thanks for your responses. The authors have addressed several questions with more detailed experiments and clarifications. And I would like to specify a few questions mentioned:
>
> * About Figure 1: I think Figure 1 just shows the operation, it needs to show more correspondence. As the author clarified "times series also require division into patches to eliminate the randomness", how does the figure convey this?
> * About Figure 2: It is hard to distinguish patches and windows by the line style. It can be more intuitive to use some color for blocks. It is better to explain the circle (time points localized in this or just sub-series in the period?) and avoid dense arrows.
> * About Table 2. As far as I know, ETT is not a well-acknowledged univariate forecasting benchmark, which contains the covariates for predicting each other. Why not test them on a more widely accepted M4 benchmark as proposed in NBEATS[1]?
>
> Considering the above responses, I am more convinced of the author's motivations. I think the contribution is enough if this work aims to show the structural similarity of time series and image and to make these two modalities benefit a unified framework (finding the same foundation model). However, it still needs more cases to support this  (as the author newly included TNT), focus more on the conclusions instead of detailed structure, and provide some takeaway messages. If this article is to be written in this way, I think there are at least a few points that need further consideration:
>
> - Time series forecasting needs not only locality. It has many properties (such as periodicity) to consider that images do not.
> - The scale of the time series and image are also different. Time series do not have a fixed range like RGB domains.
> - Not only does the attention model matter, but it is also an essential task to consider the other parts. The current projection head, which is a straightforward linear layer from the past features to the future prediction, can overfit on some of the small datasets of the paper since recent research find that a simple linear layer can especially work better on this [2].
>
> Overall, I raised the score to 3. If the author can kindly take into consideration of the experiments and presentation of the paper, I will go further on this.
>
>
>
> [1] N-BEATS: Neural basis expansion analysis for interpretable time series forecasting.
>
> [2] Revisiting Long-term Time Series Forecasting: An Investigation on Linear Mapping.

---

> > ### Author Response · Authors · 2023-11-22
> > **Response to Reviewer SUuV (2nd round, Part 1/2)**
> >
> > ### **Thanks for your timely response**
> > &emsp;&emsp;In our last response, we elaborated on our motivation and contributions to this work, which obtain your recognition. Indeed, in dealing with this kind of task across multiple modalities, we must be very careful about the rationality of the transfer. We appreciate your rigor in this regard.
> >
> > ### **About Figure 1**
> > &emsp;&emsp; Regarding the “time series also require division into patches to eliminate the randomness”, this is a factual statement, and Figure 1 does not need to be associated with it. We now understand the wording is misleading. Thanks for pointing out this. In a revised version, the words "as shown in Fig.1" before this sentence are removed. The purpose of Figure 1 is only one: make readers aware of the structural similarities between images and time series. This is the most core part of our motivation.
> >
> > ### **About Figure 2**
> > &emsp;&emsp; Thanks for mentioning this. In Figure 2, the distinction between patch and window is indeed somewhat unclear, and the arrows describing attention are also somewhat chaotic, which will not benefit reader understanding. We have made appropriate adjustments based on your suggestions.
> >
> > ### **About Table 2**
> > &emsp;&emsp; We’re afraid not. The ETT dataset is commonly used for long-term forecasting, both for univariate and multivariate forecasting. On the other hand, the M4 dataset is for short-term forecasting. Our work is mainly focused on long-term, as the title emphasizes. In recent works on univariate forecasting (especially long-term), the ETT has been the main dataset used, such as in Informer (2021), FEDformer(2022), DLinear(2023), and others.
> >
> > &emsp;&emsp; Nevertheless, extenting Swin4TS to short-term forecasting is attractive. We also evaluate the performance of Swin4TS/CI on M4 dataset, compared with recent baselines (results are from the paper of TimesNet). As shown, Swin4TS (without hyperparamters tuning, just one shot) achieves comparable performance with SOTA-method TimesNet, suggesting the essential superiority of architecture of Swin4TS. WA in this table mean weighted average on measurement of [Year, Quarter, Month and others]
> >
> > |      Models     | Swin4TS/CI （Ours） | TimesNet （2023） | N-HiTS (2022) | N-BEATS (2019) | DLinear (2023) | FEDformer (2022) |
> > |:---------------:|:-------------------:|:-----------------:|:-------------:|:--------------:|:--------------:|:----------------:|
> > | WA-SMAP         |        11.806       |       11.829      |     11.927    |     11.851     |     13.639     |      12.840      |
> > | WA-MASE         |        1.591        |       1.585       |     1.613     |      1.599     |      2.095     |       1.701      |
> > | WA-OWA          |        0.851        |       0.851       |     0.861     |      0.855     |      1.051     |       0.918      |
> >
> > ### **For the 1st concern**
> > &emsp;&emsp;Indeed, there are many differences between time series and images, as described in a newly added section in 'Conclusion'. Or, the gap you mentioned before. However, these features can be integrated into the underlying patterns and then learned by the Transformer. Our goal is to focus on the common characteristics of these two modalities and model the time series by ViTs without too much specifically designed.
> >
> > ### **For the 2nd concern**
> > &emsp;&emsp; This won't be a problem. In practical applications, time series data are normalized to stabilize neural network training, just as images also require normalization before processing.
> >
> > ### **For the 3rd concern**
> > &emsp;&emsp; Indeed, many parts of our work are related to attention. This is due to two reasons. Firstly, the window-attention is a key design of Swin Transformer to reduce computational complexity. Secondly, the attention map in the Results section is used as a tool to reflect the multivariate correlation and the effectiveness of hierarchical design. We acknowledge the importance of other modules, such as simple linear layers. In fact, DLinear is just a simple linear layer, which performs well. But there is still a significant gap between DLinear and Swin4TS, suggesting that the superiority of Swin4TS is not just due to the last linear layer.

---

> > > ### Author Response · Authors · 2023-11-22
> > > **Response to Reviewer SUuV (2nd round, Part 2/2)**
> > >
> > > ### **The overall reply**
> > > &emsp;&emsp; Overall, in this stage, you suggested us to show more case (like TNT) to support our view and change the presentation of this paper. Regarding more ViT4TS cases. Due to limited time, although we can easily design many ViT4TS architectures, we may not be able to provide experimental results for them. Please understand this. Here are several possible designs:
> > >
> > > - CrossViT4TS. The core of CrossViT (https://arxiv.org/abs/2103.14899) is to use a hierarchical structure to parallelly process local and global information by two Transformers. Then local and global information is fused through cross-attention mechanism. Obviously, this design is independent of image, so CrossViT can be easily migrated to time series modeling, that is CrossViT4TS.
> > >
> > > - DeiT4TS. DeiT (https://arxiv.org/abs/2012.12877) is also a classic model, which proposes a teacher-student distillation training strategy for ViT. This framework is also independent of image features and can be migrated to time series modeling, that is DeiT4TS.
> > >
> > > - DeepViT4TS. The core of DeepViT (https://arxiv.org/abs/2103.11886) is the Re-attention mechanism, which solves the problem of the attention tendency to be consistent when the Transformer layers go deeper. This problem is not unique to the image modality but also exists in time series. Therefore, DeepViT can also be adapted to time series after slight modification, i.e., DeepViT4TS.
> > >
> > > In fact, a lot of ViT models are independent of image features and actually focus on the common problems of Transformer models in processing sequence data. Therefore, these models can be migrated to time series.
> > >
> > > &emsp;&emsp; Regarding the presentation of this paper. In fact, **the presentation of our paper follows the style of the original ViT**. ViT first explores the structural similarity between image and word series and models them using Transformers borrowed from the NLP domain. **It also devotes considerable space to the description of ViT’s architecture, rather than being too abstract and discussing more ViT-like models**. The authors then confirm the effectiveness of ViT through sufficient experiments.
> > >
> > > &emsp;&emsp;We understand that the reviewer may prefer a brief introduction of the Swin4TS architecture, followed by detailed exploration of more ViT4TS cases. However, this may hinder readers’ understanding of the principles behind Swin4TS and their effectiveness. In our opinion, **it is crucial for readers to first fully understand Swin4TS’s principles and effectiveness, and then make them to accept another model TNT4TS to strength the understanding of this methodology.** As for attempt of more ViT4TS models (like CrossViT4TS mentioned above), it can be left to later researchers, after the validity of the technical route of ViT4TS is clearly clarified.
> > >
> > > &emsp;&emsp;Finally, **the contributions of our work are two-fold**. Firstly, we indeed propose a SOTA and efficient method Swin4TS and validate its effectiveness through extensive experiments. Secondly, the methodology of ViT4TS can motivate more attempts to facilitate the development of time series. **Overemphasizing either contribution will damage the integrity of the story.** Please kindly consider again.

---

> > > > ### Comment · Reviewer_SUuV · 2023-11-23
> > > >
> > > > I appreciate the authors' effort in answering my questions. The authors have addressed most of my concerns about the contributions and paper presentations.
> > > >
> > > > Thanks for including a detailed discussion on ViT4TS architectures, which gives more insights into the structural similarity of time series and images and how to facilitate it.
> > > >
> > > > I understand the time is limited. However, I think it is still important to compare the forecasting results on larger datasets and ablate more extensively, since the current contributions, as the author clarified, are making the forefront of ViTs to facilitate research in time series analysis. While it is important to find evidence of ViTs that work, it should also convey what can malfunction, with in-depth consideration of the modality gap. The current version does not pay much attention to this point. So the novelty may remain an issue. I believe the paper will go further and it will be tackled by the author in the future.
> > > > Overall, I raised my score to 5. Thank you once again for clarifying a lot of the points.

---

### Official Review · Reviewer_9qJL · 2023-11-04

**Soundness:** 3 good
**Presentation:** 2 fair
**Contribution:** 2 fair
**Rating:** 5
**Confidence:** 4

**Summary:**

This paper proposes to transfer the success of vision transformer to time series forecasting. The proposed method incorporates window-based attention and hierarchical representation from Swin Transformer to long-term time series forecasting. Experimental results showed state-of-the-art performance.

**Strengths:**

1. It's novel to incorporate the window-based attention and hierarchical representation from Swin Transformer to time series forecasting.

2. The performance is better than baselines.

**Weaknesses:**

1. The current time series forecasting datasets are pretty small, and performance may be satuated or over-fitting. Could the method be used for larger datasets?

2. Scalformer [1] also uses hierarchical design and the scales of time series data, this paper didn't mention and compare the similarities and differences with Scalformer.
[1] Shabani, Amin, et al. "Scaleformer: iterative multi-scale refining transformers for time series forecasting." ICLR (2023).

**Questions:**

1. The current time series forecasting datasets are pretty small, and performance may be satuated or over-fitting. Could the method be used for larger datasets?

2. Scalformer [1] also uses hierarchical design and the scales of time series data, Could this paper compare the similarities and differences with Scalformer.

[1] Shabani, Amin, et al. "Scaleformer: iterative multi-scale refining transformers for time series forecasting." ICLR (2023).

---

> ### Author Response · Authors · 2023-11-18
> **Response to reviewer 9qJL**
>
> &emsp;&emsp; First of all, thanks very much for your comments.
>
> &emsp;&emsp; Regarding the issue of dataset size, **most existing baselines for time-series prediction use the 8 datasets mentioned in the text as benchmarks. We selected them in order to have a fair comparison with other baselines**. However, as you mentioned, these datasets are relatively small and there are possible distributional shifts between the training and validation sets, which is a problem faced by all existing baseline models.
>
> &emsp;&emsp; In our collaboration project with a wind farm, we applied Swin4TS to predict wind power. The training dataset contained wind power data for 14 wind turbines, recorded every 15 minutes over a span of 5 years, comprising approximately 175k timesteps, far larger than the aforementioned 8 datasets. The goal was to predict wind power for the next 24 hours (96 points) and 72 hours (288 points), with an input size of 288. We compared three models, PatchTST, DLinear, and LightGBM. The prediction results are shown below (the results are in terms of normalized MSE).
>
> |            |     Swin4TS/CI    |     Swin4TS/CD    |     PatchTST    |     DLinear    |     LightGBM    |
> |:------------:|:-------------------:|:-------------------:|:-----------------:|:----------------:|:-----------------:|
> |     96     |     0.472         |     0.468         |     0.512       |     0.587      |     0.502       |
> |     288    |     0.632         |     0.652         |     0.633       |     0.781      |     0.649       |
>
> &emsp;&emsp; Swin4TS has performed well in predicting 96-point and 288-point. Unfortunately, due to commercial cooperation agreements, we cannot open this dataset. **We believe that Swin4TS's superior architecture is the reason for its better performance rather than the accidental fit to specific datasets, and we expect it to perform well on other large datasets as well**.
>
> &emsp;&emsp; Thanks for reminding us of the Scaleformer. Scaleformer and Swin4TS both use a hierarchy design, but their architectures are very different. Firstly, Scaleformer uses an external hierarchy design to obtain predictions at multiple scales, while Swin4TS uses an internal hierarchy design to concentrate information of different scales in the final representation and obtain the final prediction. Secondly, the prediction module in Scaleformer is totally independent, and Scaleformer is compatible with various transformer-based models. However, Swin4TS is a complete model, making it difficult to directly compare the performance with Scaleformer. Due to length limitations, we have added a short comment on Scaleformer in “Related Work” of the revised manuscript.
>
> &emsp;&emsp; **Although you did not mention the contribution, we believe that the contributions of this work might be underestimated. We not only provide an effective time series prediction framework, but more importantly, it confirms the structural similarity of time series and image, which indicates the possibilities of modeling time series modality using architectures from image modality. We elaborated on this point in detail in our response to reviewer 9ZZq and SUuV. We also supplemented the design of a TNT4TS framework to support our view. We sincerely hope you can consider our work again.**

---

### Official Review · Reviewer_9ZZq · 2023-11-06

**Soundness:** 3 good
**Presentation:** 3 good
**Contribution:** 2 fair
**Rating:** 5
**Confidence:** 4

**Summary:**

The paper proposes use Swin Transformers (popularly used as vision transformers) for long-term time-series forecasting (ltsf). The paper proposes both channel independent and channel dependent models for multivariate LTSF forecasting. It shows that the the models can be beat or match prior transformer based baselines. More over the time-complexity does not scale quadratically with the total number of patches as in PatchTST because of the use of shifted window based attention.

**Strengths:**

1. I think the SWin transformer idea fits well into LTSF tasks because it reduces the number of self attention operations which is useful for long concepts and hierarchical representations are logically the right choice for extracting multi time-scale features from time-series data.

2. For the CD model, the SWin transformer concept can again be applied to have not so expensive cross series attention although I have some questions in this regard.

3. The time-complexity is understandably better than other transformer based models.

4. The performance on the model is more often than not better than PatchTST which is the next best transformer based model.

4. The paper is easy to read.

**Weaknesses:**

1. The paper does not compare to non-transformer based baselines which are computationally more efficient.

2. The ordering of the series should matter for the CD model which not a desirable property.

3. The last layer of the CD model is not fully clear.

4. The model performance is not monotonically increasing with size of context in some corner cases which is not the case with other SOTA models.

5. Is the ablation of varying hierarchical design done in the correct manner? As I understand more hierarchies also mean more layers? Is it not better to keep the number of layers the same and just have hierarchy in one model while the scales are fixed in all layers in the ablation one?

6. What would be the strategy to handle static and dynamic covariates ?

Overall I believe the paper is well written and the use SWin is logical at least in the CI case. The architecture is not novel but nevertheless has not been used for time-series before. I would like to see the answers of my questions before deciding on the final score. It would also be good if the authors can think of some way to show whether the different scales in the hierarchy extract different types of information like long term trends vs local seasonality.

**Questions:**

1. Can the authors elaborate what the last layer looks like for the CD model? is it just mapping the flattened version of all output tokens to the future of all M time-series. In this case is this layer not too big? There might be better designs that map only specific patches to specific output predictions.

2. For vision models contiguous patches in the window are visually close in the image. This is not the case for the channel dimension in the CD time-series case. In other words does the order of the time-series matter? It should not matter in the ideal case. Can the authors show this experimentally?

3. Since the paper has a section on time-complexity it would be wise to compare with non transformer methods like TiDE (https://arxiv.org/pdf/2304.08424.pdf) and N-HiTS (https://arxiv.org/pdf/2201.12886.pdf). These methods are much faster and have comparable or better performance. For instance the numbers reported in the traffic dataset for the TiDE model are better than the ones reported in the paper. It would be good to cite and compare to these MLP based works.

4. It seems that for some horizon lengths in Figure 6 the performance decreases a little bit with increasing context. This does not seem to be the case for PatchTST and TiDE. Any intuition on why this might be happening?

---

> ### Author Response · Authors · 2023-11-18
> **Response to reviewer 9ZZq (Part 1/2)**
>
> &emsp;&emsp; Thanks very much for your review and suggestions. We next reply to your questions and concerns point to point.
>
> ### **For the 1st concern about weaknesses**
> &emsp;&emsp;**In fact, we have compared three non-transformer methods:** DLinear (MLP-based), TimesNet (CNN-based), and MICN. Please see the description in the “Compare baseline” section.
>
>  ### **For the 2nd concern**
>  &emsp;&emsp;Indeed, under the CD strategy, multiple time series can not be treated as an image, as the order of the time series can vary. In our code implementation, the order of the time series is shuffled in each training batch, and we found that this trick yielded better results than fixed initial order. We provided additional clarification on this point in the main text, and relevant experiments presented in the Appendix C.3.
>
> ### **For the 3rd concern**
> &emsp;&emsp; The final layer of Swin4TS/CD is directly mapped to MT (M is the number of series, and T is the prediction length) after learning a representation using Swin Transformer. **As you might know, this is not a good choice because for the 720-prediction task on the Traffic dataset, the size of final layer would be 862x720. This is also why the prediction performance on Traffic (M=862) and Electricity (M=321) is relatively poor under the CD strategy.** A reasonable way to address this would be to design a U-net architecture and gradually obtain MT through upsampling on the representation. **We added a part in Appendix D to address the effectiveness of this modified framework**. However, to ensure the algorithm’s simplicity and consistency with the CI strategy, we still chose the simpler one as mentioned above.  We did not want to complicate the main architecture because based on the motivation of this work, we hoped to demonstrate the structural similarity between the two modalities of time series and image and that they can be modeled using the same framework.
>
> ### **For the 4th concern**
> &emsp;&emsp;  Our results show that prediction performance varies non-monotonically with input length. **In fact, PatchTST has also reported similar phenomena, as shown in Table 9.** For example, for the 720-prediction task on ETTh2, 336-input leads to 0.379-MSE, 720-input leads to 0.394-MSE. **Autoformer and FEDformer exhibit similar phenomena, as shown in Figure 2 of the PatchTST‘s paper**. One possible explanation is that the input data contains both underlying patterns about the time series as well as noise. Once the input length is sufficient to capture the underlying patterns, longer input will introduce more noise and harm prediction performance.
>
> ### **For the 5th concern**
> &emsp;&emsp; In our work, we gave the analysis of the hierarchy design in two parts. The first part is the Ablation Study in section 4.2 Results, where **the hierarchy design was ablated in the manner just suggested by the reviewer**, as described in the main text. The second part is the varying hierarchical design in section 4.3 Effect of Hyperparameter (we moved this part in Appendix), which is not ablation of the hierarchy design, but rather an examination of the impact of different hierarchy architectures on the model. Please note the distinction.
>
> ### **For the 6th concern**
> &emsp;&emsp; Thanks for your suggestion. We have attempted to integrate dynamic covariates (the timestamp of the samples) into the input via embedding. **Surprisingly, there was a significant improvement in predictive performance on the Traffic dataset!** However, the effect was not observed on other datasets, which may be due to the Traffic dataset’s natural periodicity. We have incorporated this trick into the code, and added some descriptions of it in the Appendix C.6, and update the prediction results in Table 1.

---

> ### Author Response · Authors · 2023-11-18
> **Response to reviewer 9ZZq (Part 2/2)**
>
> ### **For the 1st question**
> &emsp;&emsp;  We have addressed this in our previous response. **We carefully considered your suggested approach and found that it is similar to our U-net solution**. This design and relevant experimental results are supplemented in the Appendix D.
>
> ### **For the 2nd question**
> &emsp;&emsp;  We have answered this above.
>
> ### **For the 3rd question**
> &emsp;&emsp;  We appreciate your suggestions. In the revised manuscript, we cited these two works. **However, it‘s hard for us to directly compare these two baselines in our main results. This is because the input size L is set to 720 for TiDE, while our Swin4TS sets L=512**. It is unfair to compare with it directly. In N-HiTS, the input size is varied among datasets, which makes the comparison even more difficult. We hope the reviewer can understand this. In addition, after the use of dynamic covariates as you suggested, the prediction performance of Traffic of Swin4TS is comparable with TiDE.
>
> ### **For the 4th question**
> &emsp;&emsp;  We have addressed this above.
>
> ### **For your suggestion**
> &emsp;&emsp; In addition, we analyze the information extracted by the hierarchical design at different scales. Please see Figure 6 and relevant descriptions.
>
> &emsp;&emsp; **In the end, we hope the reviewer can reconsider the contribution of our work. We not just propose an effective model to handle time series forecasting problem. More importantly, we consider the structural similarity between image (partition into patch series) and time series, which makes these two modalities can be modeled in a unified architecture. Further, time series modeling can benefit from the development of ViTs, please see the extensive discussion with Reviewer SUuV. We added some disscusions at the end of the revised manuscript, and additionlly designed a TNT4TS model to support our view, and showed relevant experimental results in Appendix E.**

---

> > ### Comment · Reviewer_9ZZq · 2023-11-20
> > **Thanks for the discussion**
> >
> > __Comparison to MLP methods:__ I would like to point out the DLinear is not an MLP as there is no non-linearity. Moreover TimesNET is also CNN based, not an MLP. Therefore I still think comparison to TiDE and NHiTS is essential. In particular Table 2 in TiDE paper has numbers from both the models. Ideally it would be good to your numbers to this table and you are free to tune the prediction horizon per dataset. Note that you already have the results over a variety of context lengths in Table 13, so you could have easily compared the best numbers. I am still confused as why this would have been unfair. Also a comparison wrt to computational complexity is warranted.
> >
> > Thanks for adding the dynamic covariate results and I am glad to see the performance boost on traffic.
> >
> > __U-Net:__ The U-Net experiments that were added was interesting. I am curious if there was a dataset where CD + U-NET performed better than CI? If that is not the case, then the practical implication of this still remains unexplored.
> >
> > __Motivation:__ If the main motivation of the paper is indeed merging image and time-series modality, then even preliminary experiments using a mixture of two modalities is warranted. My view of the paper was that it was trying to show that Swin transformers could be SOTA in long-term forecasting, which is slightly different from the motivation listed in the rebuttal answers. I tend to agree with some of the points mentioned by reviewer SUuV in this regard.
> >
> > I would like to keep the score at 5 for now but will definitely consider raising it after further discussion, including ones with other reviewers.

---

> > > ### Author Response · Authors · 2023-11-22
> > > **Response to reviewer 9ZZq (2nd round, Part 1/2)**
> > >
> > > ### **Thanks for your timely response**
> > >
> > > ### **For the concern on MLP methods**
> > > &emsp;&emsp;Considering DLinear as MLP-based is just a personal view, because DLinear only uses one Linear Layer, which can be regarded as the simplest case of MLP. Anyway, there is no statement about 'DLinear is MLP-based' in the main text. In addition, in the last response, we said that TimesNet is CNN-based, and did not say that it is MLP-based.
> > >
> > > &emsp;&emsp;For why we said it was unfair to directly compare TiDE. In our view, a fair way to compare baselines is to evaluate all baselines on all input sizes, and then select the best result. In this way, if our result is still better than the baseline, we can be more certain that this superiority is due to differences in model architecture rather than input size. Swin4TS used an input size of 512, while the baselines we compared to were based on 96, 336, or 512 inputs, respectively. Therefore, **when we directly compare to TiDE (input size 720), we need to evaluate all the latest baselines (MICN, Crossformer, TimesNet) on input size 720, and evaluate TiDE on input size [96,336,512], then select the best result. It is difficult for us to do this within a limited time.**
> > >
> > > &emsp;&emsp;In addition, we notice that TiDE first released on arXiv on April 17th and last updated in August. Our work is finished in July, **so we think that Swin4TS and TiDE are works from the same period. Although the overall performance of Swin4TS is better than TiDE’s, direct comparison would reduce the superiority and presentation of Swin4TS**. In fact, we have examined related works this year and found that there are almost no works that have direct comparisons with TiDE. Please kindly understand it. **Alternatively, we additionly compared with N-HiTS in Table 1 and Table 8 although it is less rigorous according the discussion above.**
> > >
> > > &emsp;&emsp;Finally, We are afraid that it is unnecessary to compare the complexity of Swin4TS and TiDE or N-HiTS (MLP-based), as in pratice these models are obviously much efficient than Swin4TS.
> > >
> > > ### **For the U-net**
> > > &emsp;&emsp;U-net was only proposed as a supplement to alleviate the defective design of the last layer of Swin4TS/CD, and it has not been thoroughly discussed. If we want to further discuss its role, we may need to add the experiments of Swin4TS/CI-Unet and perform cross-validation with Swin4TS/CD-Unet on all datasets. However, this is a bit off topic for this work.

---

> > > > ### Author Response · Authors · 2023-11-22
> > > > **Response to reviewer 9ZZq (2nd round, Part 2/2)**
> > > >
> > > > ### **For the motivation**
> > > > &emsp;&emsp;Regarding your first question about why we didn’t conduct experiments that merge the two modalities.  **Our main motivation was not to merge these two modalities.** We suspect that the wording ‘within a unified framework’ or '... motivation for bridging these two domains...of a generalized multimodal model' in Introdution may have caused the reviewer to have such a misunderstanding. In a revised version, we deleted these statements. **Instead, we considered the similarity between these two modalities and suggested that they can be modeled using a same framework, which means the modeling of time series can use the ViT architectures for processing image by simply reducing the data dimension to 1-D.** Please note the difference.
> > > >
> > > > &emsp;&emsp;A briefly review of the original proposal of ViT maybe helpful to understand our motivation and contribution. Prior to ViT, the mainstream for handling CV tasks was CNN-based methods. Later, researchers attempted to split images into patch series, **also considering the structural similarity with word series, and modeled them using Transformers from NLP, thus obtaining the Vision Transformer (ViT)**. After ViT was proposed, the direction of CV research was thoroughly changed. In the original ViT paper, there was also no consideration of the mixture of the image modality and the language modality. Research into multimodal analysis cames in later works.
> > > >
> > > > &emsp;&emsp;Back to our work, **we have adopted a similar methodology by considering the structural similarity between time series and image data**, and incorporating advanced ViTs into the modeling of time series data, hence ViT4TS. Similarly, we have validated the feasibility of this assumption by Swin4TS (and TNT4TS) with detailed experiments. **Our contributions are multi-fold. Firstly, as you mentioned, we indeed proposed a SOTA method for time series forecasting. But beyond this, our work may offer a new routine to time series research (just as ViT initially offered a new routine to CV research).** It can motivate more attempts to use advanced ViTs (such as CrossViT, DeepViT, DeiT etc) to time series tasks, including but not limited to prediction tasks such as anomaly detection, classification, imputation, etc., which will greatly promote the development of time series analysis. Besides, our work has also paved the way for future research into multimodal analysis of time series and image data (or and language data further).
> > > >
> > > > &emsp;&emsp;In the first round of response to reviewer SUuV, we answered all questions and carefully elaborated on our motivations. **In the new response, reviewer SUuV has accepted our point of view and raised the score.** We suggest the reviewer read our in-depth discussion with the reviewer SUuV (although it is a bit lengthy), which may help to understand our motivation.

---

### Official Review · Reviewer_xaLq · 2023-11-07

**Soundness:** 4 excellent
**Presentation:** 4 excellent
**Contribution:** 3 good
**Rating:** 6
**Confidence:** 3

**Summary:**

This paper adapts the idea of using Vision Transformers (ViT) to the problem of time series forecasting. Vision transformers divide an image into patches and perform attention on the patches in order to attend to the whole image and yield a prediction. The idea is adapted to the problem of multivariate time series forecasting where the time series are divided into multiple windows and the windows are further divided into multiple patches. Attention layers are applied to the patches within a single window known as window attention. Shifted window attention is window attention on the time series shifted by half a window. The final architecture is a sequence of window and shifted window attentions for processing a particular scale. The model processes at multiple scales by feeding the downsampled output from the output of scale K as the input for scale K+1. The final predictions are derived using a linear layer on the flattened output from each of the scales.

Experimental results show superior performance compared to other neural network based approaches including transformer based approaches.

**Strengths:**

The main contribution of this paper is an approach for long term forecasting of time series using a transformer based approach.
- The proposed method can efficiently scale to very long time series due to a ViT like architecture dividing the time series into multiple windows and patches.
- The model is able to attend to various scales in the time series by virtue of the downscaling steps between different scale blocks.
- The CD variant of the proposed method can efficiently model the correlation between multiple time series.
- Experimental results show improved performance for long term prediction problems.

As summarized in table 4, the proposed model is computationally and memory efficient compared to the other approaches, while being able to model the correlations between multiple time series.

**Weaknesses:**

The paper does not seem to have any significant weaknesses. Some minor issues:
- Novelty: The main idea of the paper is derived from the existing Swin transformer idea, which slightly weakens the novelty of the paper.
- The confidence intervals are missing from the MSE and MAE values in the results tables.

**Questions:**

What is the strategy for selecting the hyper-parameters of the architecture? Were the hyper-parameters selected for each dataset individually using the validation set? Or were the hyper-parameters selected globally for all the problems simultaneously?

---

> ### Author Response · Authors · 2023-11-18
> **Response to reviewer xaLq**
>
> &emsp;&emsp; Thank you very much for your recognition of our work.
>
> &emsp;&emsp; Regarding the issue of the confidence interval for MSE and MAE that you mentioned, our main experiment setup uses a fixed random number, which is conventionally set to the number of current year. Therefore, MSE and MAE are the results of a single experiment, and there is no need to calculate the confidence interval. Of course, we have studied the impact of random seed (i.e. different initialization) on the model's performance, and the relevant results are in Appendix C.1. Since the results are presented in histogram form, there is no calculation of the confidence interval, please understand.
>
> &emsp;&emsp; For the selection of hyperparameters. We make similar datasets adopt the same hyperparameters, such as the 4 ETT datasets. Furthermore, different datasets may require different hierarchical designs, and the setting of hyper-parameters should ensure that the input length can be divisible during the downsampling process. In addition, other hyperparameters will reference the settings of other models, such as learning rate or early stop. Different datasets, due to their own properties, usually have different parameter sets. For example, in Traffic and Weather, the former’s periodicity is more obvious than the latter, and the interaction between multiple variables is also different.
>
> &emsp;&emsp; **For the novelty**. We borrowed the idea of Swin Transformer for modeling time series, and indeed, the novelty is not particularly strong if only considering this. **However, our contribution is not only proposing an effective method for modeling time series. More importantly, we consider the structural similarity between image (partitioned into patch series) and time series, which indicates the possibilities of modeling time series modality using architectures from image modality. We additionally supplemented a TNT4TS framework to support our view.** The reviewer can refer the detailed response to Reviewer 9ZZq and SUuV, and we hope the reviewer can reconsider the contribution of our work.

---

### Author Response · Authors · 2023-11-18
**Response to all reviewers**

&emsp;&emsp; First of all, I would like to express my sincere gratitude to the four reviewers for their comments, which have greatly contributed to the improvement of this work. Here we review the contributions of this work. We proposed the Swin4TS algorithm, which has lower complexity and is compatible with CI/CD strategies. The latter can explicitly consider the correlation between multiple variables, which has been recognized by reviewers.

&emsp;&emsp; However, in addition, we would like to emphasize that the successful application of the Swin Transformer to time series has confirmed the consistency in structure between images (partitioned into patch series) and time series. The only difference is that the former is 2D and the latter is 1D. This means that the Swin Transformer architecture is independent of CV tasks. It can not only handle time series (1D), but also image (2D), video (3D), and time-varying physical data (4D, such as weather data) within a unified framework. On the other hand, many ViTs are actually also independent of CV tasks, which means that they can be used for time series modeling. For example, we have attempted the TNT (Transformer in Transformer, an early work in ViTs) model for time series forecasting, which also performs well.

&emsp;&emsp; **In short, the significance of Swin4TS lies not only in providing a new method for handling time series prediction, but more importantly, it confirms the structural similarity of time series and image, which indicates the possibilities of modeling time
series modality using architectures from image modality. Meanwhile, it also breaks the inherent perception that Swin, TNT, and other advanced ViTs are only suitable for CV tasks. We hope that the reviewers can reconsider the contributions of this work in this regard.**

&emsp;&emsp; According to the feedback from the four reviewers, we have made the following revisions (**colored in blue**) to the manuscript:

1. For the concerns raised by reviewer 9zzq.

    - Regarding the order of multi-series of Swin4TS/CD, we supplemented an experiment in Appendix C.3 to test the effects of the ordering of multi-series.

    - Regarding the final layer of Swin4TS/CD, we have tested a more reasonable framework. However, in order to maintain consistency between the Swin4TS/CI and Swin4TS/CD architectures, this improved framework and experimental results occurred in the Appendix D instead of the main text.

    - Regarding the effect of covariates, we designed a strategy to fuse the dynamic covariates (timestamps) into the input series and found it could greatly improve the performance on Traffic dataset. Relevant results on all 8 datasets are added in the Appendix C.6.

   -  Regarding the analysis of hierarchical design, we showed the information at different scales through the attention map at different stages, which reflects the effectiveness of the hierarchical design. Please see Figure 6 in the revised manuscript.

   -  We added a baseline N-HiTS for comparison in Table 1 and Table 8.


2.  According to the suggestions of reviewer 9qJL, we added a short comment to Scaleformer in “Related Work” of the revised manuscript.

3.  According to the suggestions of reviewer SUuV, we have made appropriate adjustments in Figure 2  for better understanding.

4. We additionally cited Scaleformer and N-HiTS as the reviewers suggested.

5.  We added a short discussion in the Conclusion section to further comprehensively address our contribution.

6. Finally, we supplemented a new architecture TNT4TS in the Appendix to further support our view that the modeling of time series modality can use the same architecture as image modality.

---

### Meta-Review · Area_Chair_78y9 · 2023-12-07

**Metareview:**

The paper proposes the use of Vision Transformers for the long-term time series forecasting problem. It formulates a channel-independent and channel-dependent architecture using Swin Transformers, and showcase an improvement on state-of-the-art results on the standard Informer LTSF datasets, while avoiding the quadratic-time complexity penalty of standard transformer architectures.

The paper has interesting contributions with strong empirical results. Reviewers however expressed some concerns about lack of enough empirical exploration of the similarities and differences/failure-cases when moving from an image modality to temporal modality. There were also questions raised about a lack of more detailed experiments and ablation studies on more challenging public forecasting datasets (such as M5, Favorita, etc), and about missing comparisons against more recent MLP-based methods.

The findings in this paper nevertheless are quite relevant, and I would urge the authors to consider strengthening the paper with a more detailed experimental evaluation, and a more empirical investigation of the modality differences between the image and temporal domains

**Justification For Why Not Higher Score:**

This  paper has nice contributions, but the main concern here (which reviewers pointed out, and I tend to somewhat agree with) is around a lack of more extensive evaluation, including evaluation on larger/complex public datasets, and more empirical investigation of the modality differences between the image and temporal domains. On the whole however, I think this is more of a borderline paper than the scores suggest, and I wouldnt be strongly opposed to accepting this paper

**Justification For Why Not Lower Score:**

N/A

---

### Decision · Program_Chairs · 2024-01-16

Reject